# S-System, Geometry, Learning, and Optimization: A Theory of Neural Networks

## Abstract

We present a formal measure-theoretical theory of neural networks (NN) built on *probability coupling theory*. Particularly, we present an algorithm framework, Hierarchical Measure Group and Approximate System (HMGAS), nicknamed S-System, of which NNs are special cases. In addition to many other results, the framework enables us to prove that 1) NNs implement *renormalization group (RG)* using information geometry, which points out that the large scale property to renormalize is dual Bregman divergence and completes the analog between NNs and RG; 2) and under a set of *realistic* boundedness and diversity conditions, for *large size nonlinear deep* NNs with a class of losses, including the hinge loss, all local minima are global minima with zero loss errors, using random matrix theory.

## 1 Introduction

The recent development in the algorithm family of neural networks (NN) (LeCun et al. (2015)) that aim to solve high dimensional perception problems, has led to results that sometimes outperform humans in particular datasets, e.g., vision (He et al. (2015)). It is a computational imitation of biological NNs (Rosenblatt (1958) Fukushima (1980) Ruineihart et al. (1986)). From a theoretical view, however, it has arguably progressed for six decades as a blackbox function approximator. Researchers of various background have been intrigued by theoretical understanding of NNs. From the perspective of physics, we have Mehta & Schwab (2014) Lin & Tegmark (2017) Choromanska et al. (2015a); from that of applied mathematics, we have Mallat (2016); from that of information theory, we have Amari (1995) Shwartz-Ziv & Tishby (2017); from that of theoretical computer science, we have Arora et al. (2014); and from the machine learning perspective, we have Anselmi et al. (2015) Anselmi et al. (2016) Jeffrey Pennington (2017) Ankit B. Patel et al. (2016) etc.

This work is also motivated for a better theoretical understanding of NNs. Our investigation in this work has led to a measure-theoretical theory of NNs. To the best of our knowledge, we do not find works that are intimately close to ours, and *a very detailed discussion on related works is written in appendix H*. Our main contributions are summarized as follows.

- Built on the formalism of probability coupling theory, we derive an algorithm framework, named Hierarchical Measure Group and Approximate System (HMGAS), nicknamed S-System, that is designed to learn the complex hierarchical, statistical dependency in the physical world, of which the hierarchical structure is formulated as measure-theoretical assumptions.
- We show that NNs are special cases of S-System when the probability kernels assume certain exponential family distributions. Activation Functions are derived formally. We further endow geometry on NNs through information geometry, and quantitatively show NNs implement Renormalization Group (RG), of which the large scale property to be renormalized is dual Bregman divergence, or informally semantic difference, and completes the analog between NNs and RG.
- S-System shows NNs are inherently stochastic, and under a set of *realistic* boundedness and diversity conditions, it enables us to prove that for *large size nonlinear deep* NNs with a class of losses, including the hinge loss, all local minima are global minima with zero loss errors, and regions around the minima are flat basins where all eigenvalues of Hessians are concentrated around zero. This part is almost a complete work on its own right, and most of the contents have to be put in the appendices. This unusual fact is because we feels like we need to solve a known hard problem to convince ourselves and the community the plausibility of S-System proposed.

We summarize insights of our theoretical results as follows.

- **Inherent stochasticity of NNs and renormalization/coarse-graining on semantic difference/geometry on stochastic manifolds of NNs (summary of section 4).** Activations in the intermediate layers are function of estimated realization of random variables, of which the true probability measure is transported from the input data. Activation functions used in practice actually assume the exponential family distributions, which explains why NNs learn templates, since the mean of an exponential family distribution uniquely determines its distribution. Forward propagation is maximization of expected data likelihood. A stochastic manifold structure is endowed on the intermediate space of NNs, where the "distance" is defined to characterize the semantic difference between events/samples. *As the layers go deeper, the semantic difference potentially becomes gradually coarse-grained to reflect the higher level semantic difference, e.g., dogs vs cats, while ignoring lower level variations, e.g., textures (theorem 4.1).* This is true due to the information monotony phenomenon: by blocking half of the information from propagating (using ReLU as an example), information related to irrelevant variations could potentially be discarded. The semantic difference is the physical quantity to renormalize to produce large scale properties, which is missing in the incomplete analog between NNs and RG (Mehta & Schwab (2014) Lin & Tegmark (2017)). Lastly, the event spaces of samples are the objects to study if one wants to study the geometry of NNs, e.g., symmetry in the geometry. For example, robustness to deformation in images can be characterized as a close "distance" between two event collections where one is obtained by deforming all events in the other. An example is provided in example 4.1.
- **Optimization in NNs (summary of section 5 and appendix F).** The stochasticity identified enables us to analyze NNs stochastically in its full complexity. It enables us to characterize the optimization behaviors of NNs in theorem 5.1 described in the contributions above. It explains the optimization myths of NNs that though being non-convex NNs can optimize the loss to zero, and why learning progresses slowly when approaching the minima. *Informally, a huge number of cooperative yet diverse neurons can group samples into arbitrary groups corresponding to labels.* The assumptions made in the theorem are sufficient practice-guiding preconditions instead of unrealistic assumptions made to make the proof work. It explains why centering of neuron activation (Glorot & Bengio (2010)) is helpful, for it helps to let the eigenvalues of the Hessian of the risk function be symmetric w.r.t. y-axis (theorem F.2), thus guaranteeing the existence of negative eigenvalues to provide loss-minimizing directions; why normalization of neuron activation (Ioffe, Sergey and Szegedy (2015)) is helpful, for the boundedness and diversity conditions 5.1 5.2 ask the correlation between neurons, formulated as cumulants, to be small, and normalization of standard deviation possibly maintains the conditions throughout the training (appendix F.3); why the larger the network is, the easier for it to reach zero error, for our results on optimization is a probably-approximately-correct type result where the error is controlled by the size of the network — the underlying reason is complicated, and we refer readers to appendix F.2.
- **Functional purpose of NNs (summary of section 3).** This is the most important one, but also perhaps the hardest one to get. *S-System shows NNs work by grouping samples/events into groups — which emerges through optimizing an objective function — and estimate/approximate their true probability measure through empirical observations.* The group is a formalization of semantics, and explains how discrete labels emerge from continuous samples, i.e., a label identifies a group of events/samples. A proper implementation of the above process applied hierarchically creates an adaptive complex system that consists of a huge number of neurons. The neurons represent different groups of events of low mutual correlation, which preconditions the optimization results.

## 2 NEURAL NETWORK, A POWERFUL INFERENCE MACHINE OF PLAUSIBILITY OF EVENTS IN THE PHYSICAL WORLD

This section aims to give an intuitive description of the formal definition of NN given and its behaviors without delving into mathematical details; and it also serves as the paper outline. *The rest of the paper characterizes the informal description in this section formally.*

If an agent wants to interact with the world, for whatever reasons, it first needs to perceive it. A way to perceive the world is to measure certain physical objects of the world, which could be implemented as sensors of the agent, e.g., a camera measuring the spatial configuration of photon intensity. However, the measurement data (formalized in definition 1) record many events that happen in the same spatial and temporal span, and those events are entangled in the measurements (formalized in

assumption 3.5). *To perceive events happening in the world, e.g., a lion is nearby, a mechanism is needed to recognize it from measurement data, e.g., sensed spatial photon patterns.*

This leads to the problem of the structure of the physical world, which the still unknown mechanism at least needs to relate to if it aims to recognize events in the world well. This leads us to Complex System (Bar-Yam (1997) Nicolis & Nicolis (2012) Newman (2009)). The research in the field allows us to safely say that one of the most important structures of the physical world is hierarchy (Amderson (1972)). Atoms form molecules; molecules form organism, and inorganic material; organisms form creatures, which form ecology system; inorganic material forms planets, then solar systems, then galaxies. *Of the scales across the hierarchy of the world, the events at a lower scale interacting with a particular way form events at a higher scale.* The hierarchy in nature is formulated as assumptions 3.1 3.2 3.3 3.4 3.5.

In section 3, S-System is introduced in definition 3 to recognize and represent the hierarchy of events from measurement data, formulated measure-theoretically and based on probability coupling theory (Thorisson (2000)). S-System formalizes the idea that a creature is not going to reproduce an impartial representation of the world, it only captures the events that cater to its need, e.g., the survival need to identify lion, and within its reach and capacity, i.e., the amount of measurements it can gather. *S-System recognizes a hierarchy of events from the measurements, not exactly in the sense of physical reality — if a creature never measured/saw a black swan, it does not mean there aren't any — but in the sense of a manually created hierarchical groups of events, and each group is perhaps given a tag/name.* For examples, some low level object groups are named as edges; a higher level, named textures; a higher level, named body parts; an even higher one, named body, e.g., lion.

In section 4, a NN is shown to be an implementation of an S-system (shown in definition 4), when the measure is being approximated with compositional exponential family distributions (defined in definition 6). Activation Function is derived formally in definition 5. The geometry structure of the representation built by NNs is defined in definition 7 as stochastic manifolds, and benefits of hierarchy are shown quantitatively as the coarse graining on "distance" in theorem 4.1, which is the large scale physical quantity to renormalize in RG. The section shows that a NN is an inference system to infer how plausible groups of events forming a hierarchy have occurred.

We have formally defined NNs and given the benefits of hierarchy, but the question remains is, "is it practical?". This is addressed in section 5. The hierarchical organization of NNs to recognize and represent the hierarchical physical world has made itself a complex system. *Contrary to existing shallow models, or simple models, it is exactly the complexity built by a hierarchy that makes NNs the most powerful inference model.* A large collection of cooperative yet autonomous neurons, formalized as assumptions 5.1 5.2, gives NNs the ability to partition events into arbitrary groups and infer the plausibility of any groups of events (proved in theorem 5.1), which is the emergent behavior emerging from the disorder in the NN complex system.

Lastly, we note the notation used. All scalar functions are denoted as normal letters, e.g., $f$; bold, lowercase letters denote vectors, e.g., $\boldsymbol{x}$; bold, uppercase letters denote matrices, e.g., $\boldsymbol{W}$; normal, uppercase letters denotes random elements/variables, all the remaining symbols are defined when needed, and should be self-clear in the context. r.e. and r.e.s are short for random element, random elements respectively, so are r.v. and r.v.s for random variable and random variables. To index entries of a matrix, following Erdős et al. (2017), we denote $\mathbb{J} = [N] := \{1, \ldots, N\}$, the ordered pairs of indices by $\mathbb{I} = \mathbb{J} \times \mathbb{J}$. For $\alpha \in \mathbb{I}, A \subseteq \mathbb{I}, i \in \mathbb{J}$, given a matrix $\boldsymbol{W}$, $w_\alpha, \boldsymbol{W}_A, \boldsymbol{W}_{i:}, \boldsymbol{W}_{:i}$ denotes the entry at $\alpha$, the vector consists of entries at $A$, the $i$th row, $i$th column of $\boldsymbol{W}$ respectively. Given two matrix $\boldsymbol{A}, \boldsymbol{B}$, the curly inequality $\preceq$ between matrices, i.e., $\boldsymbol{A} \preceq \boldsymbol{B}$, means $\boldsymbol{B} - \boldsymbol{A}$ is a positive definite matrix. Similar statements apply between a matrix and a vector, and a matrix and a scalar. $\succeq$ is defined similarly. $\kappa$ denotes cumulant, whose definition and norms, e.g., $|||\kappa|||$, are reviewed at appendix F.2. $:=$ is the "define" symbol, where $x := y$ defines a new symbol $x$ by equating it with $y$. tr denotes matrix trace. $\mathrm{dg}(\boldsymbol{h})$ denotes the diagonal matrix whose diagonal is the vector $\boldsymbol{h}$.

## 3 PHYSICS, CONDITIONAL GROUPING EXTENSION AND S-SYSTEM

In this section, a mechanism, Hierarchical Measure Group and Approximate System, nicknamed S-System, to recognize and represent events in the physical world from measurements is introduced with formalism from probability coupling theory (Thorisson (2000)) in a measure-theoretical way.

### 3.1 Physical Probability Measure Space and Sensor

To begin with, we formally define the assumptions made on the physical world.

**Assumption 3.1** (Physics). *All events in the physical world makes a probability measure space* $\mathcal{W} := (\Omega, \mathcal{F}^{\mathcal{W}}, \mu^{\mathcal{W}})$, *where* $\Omega$ *denotes the event space,* $\mathcal{F}^{\mathcal{W}}$ *is the* $\sigma$-algebra on $\Omega$; $\mu^{\mathcal{W}}$ *is the probability measure on* $\Omega$. *We call* $\mathcal{W}$ **Physical Probability Measure Space (PPMS)**. *To avoid confusion, we note that* $\Omega$ *denotes the event space of PPMS throughout the paper.*

**Assumption 3.2** (Hierarchy). $\Omega$ *has a hierarchical structure, which means* $\Omega = \cup_{s \in \mathcal{S}} \Omega_s$, *where* $s$ *is named* **scaled parameter**, $\mathcal{S}$ *is a* **poset**, *i.e., a set with a partial order,* $\Omega_s$ *are event spaces, and for* $s, s' \in \mathcal{S}, s < s', \Omega_{s'} \subseteq \sigma(\Omega_s)$, *where* $\sigma(\Omega_s)$ *is the* $\sigma$-algebra generated by $\Omega_s$.

For $s, s' \in \mathcal{S}, s < s'$, we say $\Omega_{s'}$ is *composed* by $\Omega_s$. Furthermore, when $\omega_{s'} \in \sigma(\cup_{s \in I_s} \{\omega_s\})$, where $I_s$ is an index set and for any $s \in I_s, \omega_s \in \Omega_s$, we say $\omega_{s'}$ is composed by $\omega_s$. As motivated in section 2, to perceive the events happening in the world, measurements need to be collected, which is formalized as a r.e..

**Definition 1** (Measurement Collection). *A measurement collection is a random function* $X$ *that supported on PPMS* $\mathcal{W}$ *with an induced probability measure space* $\mathcal{X} := (\Omega^{\mathcal{X}}, \mathcal{E}^{\mathcal{X}}, \mu^{\mathcal{X}})$, *where* $\Omega^{\mathcal{X}} := \{x | x : \mathbb{U} \to \mathbb{V}\}$ *and* $\mathbb{U}, \mathbb{V}$ *are unspecified the domain and codomain.*

We make the following assumptions on $X$. It characterizes the capability and limitation of a sensor and the phenomenon that for an event $\omega_{s'}$ composed by a lower scale event $\omega_s$, the time/place/support where $\omega_{s'}$ happens contains that of $\omega_s$.

**Assumption 3.3** (Resolution). *For any measurement collections, a lower bound of scale parameter* $s_0$ *exists, such that* $\forall s \in \mathcal{S}$, $s$ *is comparable with* $s_0$, *and* $\forall \omega_s \in \Omega, s < s_0, \mu^{\mathcal{X}}(X(\omega_s)) = 0$. *We call* $\Omega_{s_0}$ **the events of the lowest measurable scale**.

**Assumption 3.4** (Measurability). *measurements are physical, i.e.,* $\mathcal{E}^{\mathcal{X}} \subseteq \sigma(\Omega_{s_0})$.

**Assumption 3.5** (Containment). *Given any two comparable scale parameter* $s, s', s < s', \omega_s \in \Omega_s, \omega_{s'} \in \Omega_{s'}$, *where* $\omega_{s'}$ *is composed by* $\omega_s$, *we have* $supp(X(\omega_s)) \subseteq supp(X(\omega_{s'}))$, *where supp denotes the support of* $X(\omega) \in E$, *i.e., the domain of* $X(\omega)$ *where* $X(w) \neq 0$ *(we assume the zero element is defined, and indicates nothing has been measured).*

The containment phenomenon is troublesome, along with the phenomenon that it is possible for any events $\omega, \omega' \in \Omega$ to have overlapping support $supp(X(\omega)), supp(X(\omega'))$, even they do not have any composition relationship. This is an inherent problem of measuring: it has collapsed all the events across scales and within the same scale in the same measurement units of the sensor, e.g., pixels at the same location in the image sensor. To perceive certain event has occurred from $X$, a mechanism is needed to disentangle it from other events.

### 3.2 S-System: Hierarchical Measure Group and Approximate System

In this subsection, we introduce Hierarchical Measure Group and Approximate System, nicknamed as S-System. Following the motivation described of S-System in section 2, we create extensions of the probability measurable space $\mathcal{X}$ that "reproduce" measure of higher scale events. The extensions are created hierarchically, by Conditional Grouping Extension (CGE). For a review of the coupling theory and probability measure space extension, please refer to appendix A.

**Definition 2** (Event Representation; (Partial) Conditional Grouping Extension). *Let* $T$ *be a r.e. in measurable space* $(E, \mathcal{E})$ *defined on a probability space* $(F, \mathcal{F}, \mu)$, *a* **Conditional Grouping Extension (CGE)** *of* $T$ *is created as the following by conditioning extension and splitting extension.*

*First, a conditioning extension* $(\Omega^{\hat{\mathcal{H}}^e}, \hat{\mathcal{H}}^e, \mu^{\hat{\mathcal{H}}})$ *of* $T$ *is created with* $((E, \mathcal{E}), (\Omega^{\hat{\mathcal{H}}}, \hat{\mathcal{H}}))$ *probability kernel* $\hat{Q}(\cdot, \cdot)$, *of which an external r.e.* $\hat{H}$ *in measurable space* $(\Omega^{\hat{\mathcal{H}}}, \hat{\mathcal{H}})$ *is created with law*

$$\mu^{\hat{\mathcal{H}}}(\hat{H}|T) = \hat{Q}(T, \hat{H})$$

*Then a splitting extension* $(\Omega^{\mathcal{H}^e}, \mathcal{H}^e, \mu^{\mathcal{H}})$ *of* $(\Omega^{\hat{\mathcal{H}}^e}, \hat{\mathcal{H}}^e, \mu^{\hat{\mathcal{H}}})$ *is created with a* $((E, \mathcal{E}) \otimes (\Omega^{\mathcal{H}}, \mathcal{H}), (\Omega^{\hat{\mathcal{H}}}, \hat{\mathcal{H}}))$ *probability kernel* $Q(\cdot, \cdot)$ *to support an external random element* $H$ *in measurable space* $(\Omega^{\mathcal{H}}, \mathcal{H})$ *with law* $\nu$, *of which*

$$\mu^{\mathcal{H}}(\hat{H}|H, T) = Q((T, H), \hat{H}), \text{ and } \mu^{\hat{\mathcal{H}}}(\hat{H}|T) = \int Q((T, H), \hat{H})\nu(dH; T)$$

*We assume that $\hat{Q}$ is a kernel parameterized by $W(T;\boldsymbol{\theta})$, a transport map $W$ applied on $T$ parameterized by $\boldsymbol{\theta}$. The extension is well defined due to Thorisson (2000) Theorem 5.1.*

*Let $\mathcal{M} := ((\Omega^{\mathcal{H}^e}, \mathcal{H}^e, \mu^{\mathcal{H}}), \{H, \hat{H}, T\})$, we call $\mathcal{M}$ the event representation built on $T$ through a CGE — we define formally an* **event representation** *is a pair, of which the first element is a probability measure space, and the second element is a set of r.e.s supported on the space, called* **random element set** *of $\mathcal{M}$. When absence of confusion, we just call $\mathcal{M}$ the event representation built on $T$. $T$ is called the* **input random element** *of $\mathcal{M}$; $H$ the* **group indicator random element***; $\hat{H}$ the* **coupled random element***; $(H, \hat{H})$ the* **output random elements** *when we would like to refer to them in bunk; $\hat{Q}$ the* **coupling probability kernel***; $Q$ the* **group coupling probability kernel***; $(\Omega^{\mathcal{H}^e}, \mathcal{H}^e, \mu^{\mathcal{H}})$ the* **coupled probability measure space***; $\mu^{\mathcal{H}}$ the* **coupled probability measure***; $\nu$ the* **conditional group indicator measure***; $W(T;\boldsymbol{\theta})$ the* **transport map** *of $\mathcal{M}$. Given an $\omega \in \Omega^{\mathcal{H}}$, we say $W^{-1}(\omega) \subset E$ is an* **event represented/indexed/grouped by** *$H$. Since CGE will be used recursively later, to emphasize, when $\mathcal{M}$ only builds on a subset of output r.e.s of another event representation, to emphasize, $\mathcal{M}$ is called an event representation built by a* **Partial CGE***.*

We explain why they are named as Conditional Grouping Extension and Event Representation. By assumption, $\hat{Q}$ is a probability kernel parameterized by $W(T)$, e.g., the exponential family probability kernel $e^{\boldsymbol{w}^T \boldsymbol{x} - \psi(\boldsymbol{w})}$, where $T := X, W(T) := \boldsymbol{w}^T X - \psi(\boldsymbol{w})$. Suppose $X$, a measurement collection r.e., is supported by PPMS $\mathcal{W}$, from definition 1. A transport map $W$ applied on $X$ is a deterministic coupling $(X, W(X))$ that transports the measure $\mu(A)$ of an event $A \in \mathcal{E}^{\mathcal{X}}$ to $W(A)$, of which $W(X)$ is a r.e. on a measurable space $(\Omega^{\hat{\mathcal{H}}}, \hat{\mathcal{H}})$ with law $\mu^{\hat{\mathcal{H}}}$ supported on $\mathcal{W}$ where

$$\mu^{\hat{\mathcal{H}}}(W(A)) = \mu^{\mathcal{W}}(X^{-1}(A)), A \in \mathcal{E}^{\mathcal{X}}, X^{-1}(A) \in \mathcal{F}^{\mathcal{W}}$$

*That is to say $A$ is an event that are happening in the physical world, and is being measured by $X$.* The goal of S-System is to estimate the plausibility of the event $A$. However, the problem is that we do not know $\mu^{\mathcal{W}}$ (that's not to say we do not have an estimation of $\mu^{\mathcal{W}}$ empirically). That's why CGE is needed. *CGE hypothesizes a probability kernel $Q$ that approximates the probability $\mu(A)$ of events being measured (conditioning extension) grouped by the r.e. $H$ created by splitting extension.* Notice two key constructions to deal with two key challenges here: for the enormity of the event space of PPMS, a.k.a. $\Omega$, only events that happen along with current observation $X$ is estimated through conditioning extension; for events happening along with $X$, *probability is approximated in groups indexed by $H$ through splitting extension*, which physically could be broken down into countless smaller scale events that compose $A$ and some of the sub-events won't be estimated. The design could be understood as economic considerations, though probably it would be the only feasible solution to reasonably approximate $\mu^{\mathcal{W}}$. Then, the r.e. set of $\mathcal{M}$ is the manipulable object that directly connects with events in the physical world, and is named event representation.

Yet, one more problem is looming around: how possibly $Q$ approximates $\mu^{\mathcal{W}}(X^{-1}(A))$ reasonably? Suppose $A$ is a top scale event, by assumption 3.2, $A \in \sigma(\Omega_{s_L}) \subset \sigma(\Omega_{s_{L-1}}) \subset \ldots \subset \sigma(\Omega_{s_0})$, where $\mathcal{S}_l = \{s_l\}_{0 \leq l \leq L, l \in \mathbb{N}}$ is a finite set of scales. *Thus, to approximate $\mu^{\mathcal{W}}(W^{-1}(A))$ is to approximate the joint distribution of events that compose $A$, which could be factorized into the probabilities of events that compose $A$ and the probability of $A$ conditioning on the sub-events.* This asks to apply CGE recursively, through which we get an S-system.

**Definition 3** (Hierarchical Measure Group and Approximate System). *A Hierarchical Measure Group and Approximate System (S-System) is a mechanism to extend the probability measure space of a measurement collection r.e. recursively according to a poset structure $\mathcal{S}^{\mathcal{X}}$ as described in algorithm 1. The poset is called the* **scale poset** *of the S-system. Ultimately, it creates an event representation $\mathcal{M}^{\bar{\mathcal{W}}} := (\bar{\mathcal{W}}, \mathcal{O}^{\bar{\mathcal{W}}})$, where $\bar{\mathcal{W}} := (\bar{\Omega}, \mathcal{F}^{\bar{\mathcal{W}}}, \mu^{\bar{\mathcal{W}}})$ is the extended probability measure space built and is called* **Approximated Probability Measure Space (APMS)***, and $\mathcal{O}^{\bar{\mathcal{W}}}$ is a r.e. set indexed by elements of poset $\mathcal{S}^{\mathcal{X}}$. $\mathcal{M}^{\bar{\mathcal{W}}}$ is called the event representation built by S-System.*

## 4 NEURAL NETWORKS FROM THE FIRST PRINCIPLE AND ITS GEOMETRY

In the previous section, a mechanism S-System is introduced to transport, group and approximate probability measures that are of interest. It focuses on deriving a mechanism to recognize events through measurements from the first principle. In this section, we will show that Multiple Layer

---

**Algorithm 1** S-System. In the algorithm below, the `predecessor`(s) returns a index set $\mathbb{I}'$ that indexes elements in $\mathcal{S}^{\mathcal{X}}$ and $\forall i \in \mathbb{I}', s_i \leq s$ and `successor`(s) return a subset $\mathcal{S}'$ of $\mathcal{S}^{\mathcal{X}}$, where $\forall s_i \in \mathcal{S}', s_i \geq s$. For examples of the functions, refer to appendix B.

---

**Input:** $\mathcal{S}^{\mathcal{X}} := \{s_i\}_{i \in \mathbb{I}}$ is a poset with a minimal element $s_0$, whose elements are indexed by a set $\mathbb{I}$; $X$ is a measurement collection r.e. supported on $\mathcal{X} := (\Omega^{\mathcal{X}}, \mathcal{E}^{\mathcal{X}}, \mu^{\mathcal{X}})$, of which the events of the lowest measurable scale are $\Omega_{s_0}$

**Output:** an event representation $\mathcal{M}^{\mathcal{W}}$

    $\mathcal{M}_{s_0} \leftarrow ((\Omega^{\mathcal{H}_{s_0}}, \mathcal{H}_{s_0}, \mu^{\mathcal{H}_{s_0}}) := \mathcal{X}, \mathcal{O}_{s_0} := (X)), \mathcal{T}_{\text{out}} \leftarrow \emptyset, \mathcal{S}^{\mathcal{X}_t} \leftarrow \text{SUCCESSOR}(s_0)$

    **while** $\mathcal{S}^{\mathcal{X}_t}$ is not empty **do**

        $\mathcal{S}^{\mathcal{X}_{t'}} \leftarrow \emptyset$

        **for** $s \in \mathcal{S}^{\mathcal{X}_t}$ **do**

            $\mathbb{I}' \leftarrow \text{PREDECESSOR}(s)$

            $\mathcal{H} \leftarrow \bigotimes_{i \in \mathbb{I}'}(\Omega^{\mathcal{H}_{s_i}}, \mathcal{H}_{s_i}, \mu^{\mathcal{H}_{s_i}}), \mathcal{O} \leftarrow \cup_{i \in \mathbb{I}'} \mathcal{O}_i, \mathcal{M} \leftarrow (\mathcal{H}, \mathcal{O})$

            Build an event representation $\mathcal{M}_s := ((\Omega^{\mathcal{H}_s}, \mathcal{H}_s, \mu^{\mathcal{H}_s}), \mathcal{O} \cup \{H_s, \hat{H}_s\})$ on $\mathcal{O}$ through

conditional grouping extension that supports output r.e.s $(H_s, \hat{H}_s)$

            **if** SUCCESSOR(s) is empty **then**

                $\mathcal{T}_{\text{out}} \leftarrow \mathcal{T}_{\text{out}} \cup \{\mathcal{M}_s\}$

            **else**

                $\mathcal{S}^{\mathcal{X}_{t'}} \leftarrow \mathcal{S}^{\mathcal{X}_{t'}} \cup \text{SUCCESSOR}(s)$

            **end if**

        **end for**

        $\mathcal{S}^{\mathcal{X}_t} \leftarrow \mathcal{S}^{\mathcal{X}_{t'}}$

    **end while**

    $\mathcal{M}^{\mathcal{W}} \leftarrow (\bigotimes_{i \in \mathbb{I}''}(\Omega^{\mathcal{H}_{s_i}}, \mathcal{H}_{s_i}, \mu^{\mathcal{H}_{s_i}}), \cup_{i \in \mathbb{I}''} \mathcal{O}_i)$, where $\mathbb{I}''$ indexes all event representations now in the set $\mathcal{T}_{\text{out}}$

---

Perceptrons (MLP) (Ruineihart et al. (1986)) is an implementation of an S-system. *The derivation serves as a proof of concept, and as an example of S-System*, though we note that all existing NN architectures, e.g., Residual Network (He et al. (2016)), Convolutional Neural Network (Simard et al. (2003)), Recurrent Neural Network (Hochreiter et al. (1997)), Deep Belief Network (Hinton et al. (2006)) could be derived by using different measurable spaces, posets, probability kernels and successor, predecessor functions, along with manifold possibilities of new architectures. In the derivation, we will see classical activation functions emerging naturally. Then, we go further to endow geometry on event representations by defining the proper manifold structure on S-System using information geometry. It enables us to quantitatively prove the benefits of hierarchy that MLPs implement coarse graining that contracts the variations in the lower scale event spaces when creating higher scale event extensions, which plays the same role as RG in physics.

### 4.1 THEORETICAL DERIVATION OF ACTIVATION FUNCTIONS AND MLPs

Let the CGE in definition 3 be MLPCGE (definition 4), the $\tilde{t}$ in MLPCGE be obtained by transport map ReLU (definition 5), and the scale poset $\mathcal{S}^{\mathcal{X}}$ be a chain, i.e., a poset where all elements are comparable. By algorithm 1, we would obtain a MLP. The definitions are given in the following.

**Definition 4** (MLP Conditional Grouping Extension). *An MLP Conditional Grouping Extension (MLPCGE) is a CGE with the following measurable space and parametric forms of probability kernels*

$$(E, \mathcal{E}) = (\mathbb{D}^n, \mathcal{D}^n) \bigotimes (\mathbb{R}^n, \mathbb{B}(\mathbb{R}^n)), \nu(\boldsymbol{h}|\boldsymbol{t}) = e^{\boldsymbol{h}^T \boldsymbol{W}^T \tilde{\boldsymbol{t}}} / \sum_{\boldsymbol{h}} e^{\boldsymbol{h}^T \boldsymbol{W}^T \tilde{\boldsymbol{t}}}$$

$$\hat{Q}(T, \hat{H}) = \hat{q}(\boldsymbol{t}, \hat{\boldsymbol{h}}) := e^{\mathbf{1}^T \hat{\boldsymbol{h}}(\boldsymbol{t})} / (\int e^{\mathbf{1}^T \hat{\boldsymbol{h}}(\boldsymbol{t})} d\mu(\boldsymbol{t})) = e^{\mathbf{1}^T \boldsymbol{W}^T \tilde{\boldsymbol{t}}} / (\int e^{\mathbf{1}^T \boldsymbol{W}^T \tilde{\boldsymbol{t}}} d\tilde{\mu}(\tilde{\boldsymbol{t}}))$$

$$Q((T, H), \hat{H}) = q((\boldsymbol{t}, \boldsymbol{h}), \hat{\boldsymbol{h}}) := e^{\boldsymbol{h}^T \hat{\boldsymbol{h}}(\boldsymbol{t})} / (\int e^{\boldsymbol{h}^T \hat{\boldsymbol{h}}(\boldsymbol{t})} d\nu(\boldsymbol{h}|\boldsymbol{t}) d\mu(\boldsymbol{t}) = e^{\boldsymbol{h}^T \boldsymbol{W}^T \tilde{\boldsymbol{t}}} / (\int e^{\boldsymbol{h}^T \boldsymbol{W}^T \tilde{\boldsymbol{t}}} d\nu(\boldsymbol{h}|\boldsymbol{t}) d\tilde{\mu}(\tilde{\boldsymbol{t}}))$$

*where $\mathbb{D}^n$ is a $n$-dimensional discrete-valued field, i.e., $\{0,1\}^n$ or $\{-1,1\}^n$, $\mathcal{D}^n$ is the $\sigma$-algebra generated by $\mathbb{D}^n$, $\boldsymbol{W}$ is a matrix (in this case, the transport map $W(T; \boldsymbol{\theta})$ is the matrix $\boldsymbol{W}$ and*

*parameters $\boldsymbol{\theta}$ are $\boldsymbol{W}$), $\boldsymbol{t}, \hat{\boldsymbol{h}}, \boldsymbol{h}$ are realizable values of r.e.s $T, \hat{H}, H$, and $\tilde{\boldsymbol{t}}$ is obtained by applying a yet unspecified transport map on $\boldsymbol{t}$ — for now, it could be just taken as the output of an identity mapping and other possible forms are introduced when discussing activation functions — and $\tilde{\mu}(\tilde{\boldsymbol{t}})$ is the law on $\tilde{\boldsymbol{t}}$ induced by the law $\mu(\boldsymbol{t})$ on the input r.e. of MLPCGE. The meaning of the rest of the symbols is same with those in definition 2.*

Note that it is not possible to compute $\hat{q}(\boldsymbol{t}, \hat{\boldsymbol{h}})$, for $\mu(\boldsymbol{t})$ is unknown. However, we can compute $\nu$ faithfully! This is because $H$ is a manual creation/grouping instead of inherent events in PPMS Here, with some further reasoning, we will have the marvelous trick done by NNs, i.e., the Activation Function (AF). The key is only to build a full, or partial CGE upon r.e.s created by a previous CGE, using an estimated value of $H$. The deeper principles of the estimation are described in appendix G, which is the maximization of expected data log likelihood, and is part of the learning framework of S-System. When a full CGE is created upon output r.e.s. of a previous CGE, $\mathbb{D}$ is $\{0, 1\}^n$, and the estimation is done through expectation or maximum, we recover the currently best performing activation function Swish (Ramachandran et al. (2017)) or ReLU (Glorot et al. (2011)) respectively; when a partial CGE is created on the group indicator r.e.s, the estimation is done through expectation, and $\mathbb{D}$ is $\{0, 1\}$ or $\{1, -1\}$, we recover classical activation functions Sigmoid or Tanh respectively.

We derive ReLU as an example. The group indicator r.e.s $H$ divides the measure transported from the event space of input r.e. $T$ to the event space of $\hat{H}$ into groups. Intuitively, if $H$ divides the measure into two groups indexed by elements of $\mathbb{D}$, and we assume $1$ collects the measure corresponds to an event collection while $0$ collects the complement of the event collection (meaning the event collection does not occur), given a realization $\boldsymbol{t}$ of $T$, to recognize higher scale events composed by lower scale events represented by $H$, we would like to *estimate* what events are present in $\boldsymbol{t}$, and create *further coupling* with another CGE on the events that are present. Formally,

**Definition 5** (Rectified Linear Unit (ReLU))**.** *Let $T := (H, \hat{H})$ be r.e.s created by a CGE, an estimation $\tilde{\boldsymbol{h}}$ of a realization of the r.e. $H$ is obtained by*

$$\tilde{h}_i = \underset{h \in \mathbb{D}}{\arg \max} \, \nu(h|\boldsymbol{t}) = e^{h \boldsymbol{W}_{:i}^T \tilde{\boldsymbol{t}}} / \sum_h e^{h \boldsymbol{W}_{:i}^T \tilde{\boldsymbol{t}}}$$

*A further coupling is created by MLPCGE upon $T$, of which $\tilde{\boldsymbol{t}}$ is the estimated realization of the r.v. obtained by applying a transport map ReLU on $T$*

$$ReLU(T) := H \odot \hat{H}$$

*where $\odot$ denotes Hadamard product. Operationally, ReLU is a binary mask $\boldsymbol{h}$ applying on the outputs (preactivation) $\hat{\boldsymbol{h}}$ of the transport map $\boldsymbol{W}$.*

As can be seen, AFs is not a well defined object, which is actually a combination of operations from two stages of computation.

## 4.2 NN Manifold and Contraction Effect of Conditional Grouping Extension

In Section 3, motivated by the hierarchy assumption 3.2, we designed S-System. Here, using MLP as an example, and also a proof of concept, we quantitatively show the benefits of hierarchical grouping done in S-System by showing that irrelevant/uninterested variations in the lower scale events could be gradually dropped by repeatedly applying CGE, characterized by "shrinking distance" between events.

To characterize the distance, we need a geometry structure on event representations. We give an initial construction built on information geometry (Amari (2016)). For a review of manifold and information geometry, please refer to appendix C D. To begin with, we define

**Definition 6** (Compositional Exponential Family of Distributions)**.** *The form of* **compositional probability distribution of exponential family** *is given by the probability density function*

$$p(\boldsymbol{x}, \boldsymbol{h}; \boldsymbol{\theta}) d\boldsymbol{x} d\boldsymbol{h} = e^{(k(\boldsymbol{h}, \boldsymbol{x}; \boldsymbol{\theta}) - \psi(\boldsymbol{\theta}))} d\mu(\boldsymbol{x}) d\nu(\boldsymbol{h}), \, k(\boldsymbol{h}, \boldsymbol{x}; \boldsymbol{\theta}) = \langle \boldsymbol{f}(\boldsymbol{\theta}; \boldsymbol{h}), \boldsymbol{g}(\boldsymbol{x}) \rangle$$

*where $\boldsymbol{x}$ is realizable values of a multivariate random variable, $k$ is a function called* **compositional kernel** *that for a given $\boldsymbol{h}$, $k$ is the inner product between certain vector function $\boldsymbol{g}(\boldsymbol{x})$, called* **sufficient statistic***, (of which the component functions are linearly independent) and certain vector function $\boldsymbol{f}(\boldsymbol{\theta}; \boldsymbol{h})$, called* **composition function***, $\psi(\boldsymbol{\theta})$ is the normalization factor, and $\mu, \nu$ ares the laws on r.v. $\boldsymbol{x}, \boldsymbol{h}$, respectively.*

Conditioning on $\boldsymbol{h}$, $p(\boldsymbol{x}|\boldsymbol{h};\boldsymbol{\theta}) = e^{k(\boldsymbol{h},\boldsymbol{x};\boldsymbol{\theta})}d\mu(\boldsymbol{x})$ is of the exponential family. Actually, it is of Curved Exponential Family (Amari (1995)). The parametric form of kernel $Q$ of MLPCGE is of the compositional exponential family, where $k(\boldsymbol{h},\boldsymbol{x};\boldsymbol{\theta}) = \langle \boldsymbol{h}^T\boldsymbol{W}, \boldsymbol{x}\rangle$, $\boldsymbol{f}(\boldsymbol{\theta};\boldsymbol{h}) = \boldsymbol{h}^T\boldsymbol{W}$, $\boldsymbol{g}(\boldsymbol{x}) = \boldsymbol{x}$.

**Definition 7** (Neural Network Manifolds). *Let $\mathcal{M}$ be an event representation built on a measurement collection r.e. $X$ through an S-system. If probability kernels $Q, \hat{Q}$ of all CGE in the S-system are of the compositional exponential family, of which the composition kernel is parameterized by the CGE transport map, then the function space $M_s$ of measure $\mu_s^{\mathcal{H}}(\hat{H},T|H), s \in \mathcal{S}^{\mathcal{X}}$, where $\mathcal{S}^{\mathcal{X}}$ is the scale poset of the S-system, is a Riemannian manifold with the following properties:*

- *$M_s$ has a coordinate system $\boldsymbol{\eta}_s|_{\boldsymbol{h}} = (\eta_1, \ldots, \eta_n)$ that is the dual affine coordinate system of an exponential family distribution, where*

$$\boldsymbol{\eta}_s|_{\boldsymbol{h}} := \nabla_{\boldsymbol{f}(\boldsymbol{\theta};\boldsymbol{h})}\psi(\boldsymbol{\theta}) = \mathbb{E}[\boldsymbol{t}|\boldsymbol{h}] = \int \boldsymbol{t}\, d\mu_s^{\mathcal{H}}(\hat{H},T|H) = \int \boldsymbol{t}\, q((\boldsymbol{t},\boldsymbol{h}),\hat{\boldsymbol{h}})d\mu(\boldsymbol{t})$$

  *$\boldsymbol{\theta}$ is the parameters of the transport map $W(T;\boldsymbol{\theta})$, $\nabla_{\boldsymbol{f}(\boldsymbol{\theta};\boldsymbol{h})}$ takes derivatives w.r.t. composition function of $k$ and $\boldsymbol{t}$ is realizations of $T$. We call the coordinates **neuron coordinates**.*
- *$M_s$ has a Riemannian metric derived by the second order Taylor expansion of the dual Bregman divergence defined by*

$$D_{\psi^*}[\boldsymbol{\eta}_s'|_{\boldsymbol{h}} : \boldsymbol{\eta}_s|_{\boldsymbol{h}}] := \psi^*(\boldsymbol{\eta}_s'|_{\boldsymbol{h}}) - \psi^*(\boldsymbol{\eta}_s|_{\boldsymbol{h}}) - \nabla\psi^*(\boldsymbol{\eta}_s|_{\boldsymbol{h}})^T(\boldsymbol{\eta}_s'|_{\boldsymbol{h}} - \boldsymbol{\eta}_s|_{\boldsymbol{h}})$$

  *where $\psi^*$ is the Legendre dual of $\psi$. We call the divergence defined **neuron divergence**.*

In the above definition, the stochastic manifold is defined by conditioning on group indicator r.e.s $H$. To appreciate the definition, let's return back to MLP. Let $\mathcal{M}$ be the event representation built on a measurement collection r.e. $X$ by a MLPCGE, i.e., the measure of output r.e.s being $\mu^{\mathcal{H}} = e^{\boldsymbol{h}^T\boldsymbol{W}^T\boldsymbol{x}-\psi(\boldsymbol{W})}\mu(\boldsymbol{x})\nu(\boldsymbol{h}|\boldsymbol{x})$. When $\boldsymbol{h}$ is fixed, letting $\boldsymbol{f}_0 = \boldsymbol{h}^T\boldsymbol{W}^T$ we have $\mu^{\mathcal{H}}(\boldsymbol{x}|\boldsymbol{h}) = e^{\boldsymbol{f}_0^T\boldsymbol{x}-\psi(\boldsymbol{W})}\mu(\boldsymbol{x})$. It is known (Nielsen & Garcia (2009)) that the expectation statistics, i.e., $\boldsymbol{\eta}|_{\boldsymbol{h}} = \nabla_{\boldsymbol{f}_0}\psi(\boldsymbol{f}_0(\boldsymbol{W}))$, uniquely determines $\mu^{\mathcal{H}}(\boldsymbol{x}|\boldsymbol{h})$. It implies that given $\boldsymbol{h}$, $\mu^{\mathcal{H}}$ is a probability distribution, of which the most "salient" feature is the expectation. This explains why the visualization of NN representations tends to be templates (Mahendran & Vedaldi (2015) Zhang & Zhu (2018)), and the template based theories (Riesenhuber & Poggio (1999) Ankit B. Patel et al. (2016) Balestriero & Baraniuk (2018)) are partially right. Thus, the group indicator $\boldsymbol{h}$ represents the events of $\mathcal{M}$, of which the expectation is the representative. Let the transport map of $\mathcal{M}$ be $W : \boldsymbol{x} \mapsto \boldsymbol{h}^T\boldsymbol{W}^T\boldsymbol{x}$, $\mu^{\mathcal{H}}(\boldsymbol{x}|\boldsymbol{h})$ approximates the measure $\mu^{\mathcal{W}}(X^{-1}W^{-1}(A)), A \subseteq \mathbb{R}^n$ in PPMS. That's why instead of using the canonical coordinate of exponential family distribution, we use its dual affine coordinate. Though essentially the two coordinate systems are dual views on the same object, we define the manifold this way to characterize the fact that for a given NN, $D_{\psi^*}$ characterizes the degree of separation between two events $A, A' \in \Omega$ of which the probability measures $\mu^{\mathcal{W}}(A), \mu^{\mathcal{W}}(A')$ are transported by $W(X(A)), W(X(A'))$ and approximated by $\mu^{\mathcal{H}}$. Furthermore, the divergence is defined by conditioning reflects the fact events can be compared using multiple criteria, though to evaluate its implication more works are needed. For how the above definitions relate to classical definitions on NNs in information geometry, please refer to appendix H.4.

By a directed application of theorem 14 of Liese & Vajda (2006), which is called information monotony in Amari (2016), we have

**Theorem 4.1** (Contraction of divergence between events). *Let $\mathcal{A} := \mathcal{A}' \cup \mathcal{A}''$ be an event collection in event space $\Omega$ of PPMS consisting of two event collections, and two measurement collection r.e.s $X', X''$ are created for $\mathcal{A}', \mathcal{A}''$ respectively. Let $\mathbf{S}$ be an S-System, $\mathcal{M}', \mathcal{M}''$ be event representations built on $X', X''$ by $\mathbf{S}$ respectively, and $\mathcal{S}^{\mathcal{X}}$ be the scale poset of $\mathbf{S}$. Then $\forall s_1, s_2 \in \mathcal{S}^{\mathcal{X}}, s_1 < s_2$, we have*

$$D[\boldsymbol{\eta}_{s_1}'|_{\boldsymbol{h}_1} : \boldsymbol{\eta}_{s_1}''|_{\boldsymbol{h}_1}] \geq D[\boldsymbol{\eta}_{s_2}'|_{\boldsymbol{h}_2} : \boldsymbol{\eta}_{s_2}''|_{\boldsymbol{h}_2}]$$

*where $\boldsymbol{\eta}_{s_1}', \boldsymbol{\eta}_{s_2}'$ are the neuron coordinates at scale $s_1, s_2$ of $\mathcal{M}'$ respectively; so are $\boldsymbol{\eta}_{s_1}'', \boldsymbol{\eta}_{s_2}''$ of those of $\mathcal{M}''$; $\boldsymbol{h}_1, \boldsymbol{h}_2$ are arbitrarily fixed realizations of group indicator r.e.s at scale $s_1, s_2$ respectively.*

**Example 4.1** (Contraction of divergence induced by deformation). *Let $g$ be a diffeomorphism group, and $X' = g.X$, the deformed r.e. created by applying $g$ on a r.e. $X$. By the above theorem, for event representations created by an S-system, coordinated as $\boldsymbol{\eta}|_{\boldsymbol{h}}, \boldsymbol{\eta}'|_{\boldsymbol{h}}$, their distance is gradually contracted in term of neuron divergence. For a review of diffeomorphism group, refer to appendix C.*

The theorem has twofold significance. First, it shows that a recursive application of CGE would shrink the discrepancy between events, thus possessing the capacity to contract irrelevant variations in the events, though further characterizations are needed to give operational guidance. It completes the incomplete analog between NNs and RG (Lin & Tegmark (2017)), which lacks a physical quantity to renormalize to produce large scale properties. The physical quantity is shown as the neuron divergence between event representations, or more informally, semantic difference between samples. *We note essentially the large scale quantity is group indicator r.e.s of high scales that represent events that gradually have semantic meaning, of which the neuron divergence is a property.* We have present it this way since a quantity like distance is more concrete and easy to understand. Contrary to clear-cut physical quantities like temperature emerging in physics through RG, *a meaningful event group emerges through learning, which leads us to the next section.* Second, along with definition 7, it identifies the proper object if the geometry of NNs is to be studied. For example, to study the symmetry in the geometry, the object to investigate is the symmetry in the event space, of which the diffeomorphism group is a type of symmetry, and invariance is the mapping of events to the same neuron coordinates. This is in contrast with existing works that study symmetry by studying the equivariance (Cohen & Welling (2016) Dieleman et al. (2016)), or invariance (Anselmi et al. (2016)), or linearization of diffeomorphism (Mallat (2016)) in NNs through studying the changes induced by group actions in feature maps in the intermediate layers of a sample, which is a rather ad hoc object. The event space perhaps is the "mathematical ghost" lurking in Mallat (2016).

## 5    LEARNING AND OPTIMIZATION LANDSCAPE OF NNS AND S-SYSTEM

We have introduced a new algorithm family, i.e., S-System, to estimate probabilities, and shown MLP is an implementation of an S-system. Yet, to construct an operational theory of S-System, we still need a strategy to learn the parameters of probability kernels of S-System. In other word, how possibly can we ground the probability approximation created by coupling on "reality"? We will show that despite possessing the normally undesirable complexity, non-identifiability and singularity (Amari (2016)) properties, S-System could be marvelously powerful. Stating in a more familiar language, the problem translates to how a many latent variable model is able to learn? This is addressed in this section, and is the long standing optimization issue of NNs. We aim at investigating the principle underlying instead of proving the most general case. More specifically, when a set of boundedness and diversity conditions hold, we show that a NN can approximate the probability distribution of any binary group indicator r.e. given empirical measure of the r.e.; or in other words, for a class of losses, including the hinge loss, and a class of NNs, including MLP and CNN, we prove that all local minima of the empirical risk function are global minima with zero loss values.

The problem is formulated as the following. Let $\mathcal{M}^{\bar{\mathcal{W}}}$ be an event representation built by an S-system S on a measurement collection r.e. $Z$ supported on the PPMS $\mathcal{W}$; the measurable space $\mathcal{Z}$ and measure on $Z$ are $\mathcal{X} \times \mathcal{Y}$ and $\mu^{\mathcal{Z}}$ respectively, where $\mathcal{X} := \mathbb{R}^n, \mathcal{Y} := \{-1, 1\}$. Let the scale poset of S be a chain, symbolically represented as an integer set $\mathcal{S}^{\mathcal{X}} = \{0, \ldots, L\}, 0 < \ldots < L$, and $(H_l, \hat{H}_l), l \in \mathcal{S}^{\mathcal{X}}$ the output r.e.s of $\mathcal{M}^{\bar{\mathcal{W}}}$. A reward-penalty mechanism is introduced to give feedback on the "faithfulness" of approximated measure $\mu_L^{\mathcal{H}}$ as a discrepancy measure between $\nu(H_L | H_{\{1, \ldots, L-1\}}, X)$ and $\mu^{\mathcal{Z}}(Y | X)$, where $(X, Y) \in \mathcal{Z}$. *We can see that supervised learning actually uses the group indicator r.e. $\boldsymbol{H}_L$ to approximate the grouping of samples arbitrated by labels. In a certain way, it formalizes semantics.* The problem formulation is a part of the general learning framework of S-System described in appendix G, which also includes unsupervised learning, though we do not have space to discuss here.

The problem setting above is principally the same with the formulation of a binary supervised learning problem in statistic learning theory (SLT). For a classic-style formulation, the readers may refer to appendix E. The key insight of SLT is that instead of seeking a fully probabilistic formulation, the discrepancy can be formulated as an empirical risk that measures the discrepancy between $H_L$ and $Y$, calculated on a set of training samples $\{X_i, Y_i\}_{i=1, \ldots, m}$:

$$R(T) = \frac{1}{m} \sum_{i=1}^{m} l(T(X_i; \boldsymbol{\theta}), Y_i) = \frac{1}{m} \sum_{i=1}^{m} l(\hat{H}_L(X_i), Y_i) \tag{1}$$

where $T$ here denotes the hierarchical transport map built, $l$ is a loss function, and $\hat{H}_L(X_i)$ is the coupled r.e. built on $X$ at scale $L$.

Our goal is to investigate the fundamental principle that makes $R(T)$ tractable. To do this, we study the risk landscape by studying the Hessian of $R(T)$ of a particular class of loss functions, of which the eigenvalue spectrum dictates whether critical points of $R(T)$ are local minima, or saddle points. To motivate the class of losses, observing that

$$\frac{d^2}{d\boldsymbol{\theta}^2} l(T(\boldsymbol{x};\boldsymbol{\theta}), y) = l''(T(\boldsymbol{x};\boldsymbol{\theta}), y) \frac{d}{d\boldsymbol{x}} T(\boldsymbol{x};\boldsymbol{\theta}) \frac{d}{d\boldsymbol{x}} T(\boldsymbol{x};\boldsymbol{\theta})^T + l'(T(\boldsymbol{x};\boldsymbol{\theta}), y) \frac{d^2}{d\boldsymbol{x}} T(\boldsymbol{x};\boldsymbol{\theta}) \qquad (2)$$

We study the class $\mathcal{L}_0$ of functions $l$ and the class of NNs $T$, such that for $l \in \mathcal{L}_0$, it satisfies: 1) $l : \mathbb{R} \to \mathbb{R}^+$, when $y$ is taken as a constant; 2) $l$ is convex; 3) the second order derivatives $\frac{d^2}{d\boldsymbol{x}^2} l$ is zero; 4) $\min_x l(\boldsymbol{x}, y) = 0$, while for $T$ it satisfies: $\dim(T(\boldsymbol{x};\boldsymbol{\theta})) = 1$. The restriction allows us to study the most critical aspect of the risk function of supervised NNs by making the first term above zero, while the second term a single matrix (instead of an addition of matrices). The class of $l$ includes important loss functions like the hinge loss $\max(0, 1 - \hat{H}_L Y)$, and the absolute loss $|\hat{H}_L - Y|$, which were studied in Choromanska et al. (2015a) under unrealistic assumptions. The class of $T$ is the NN with a single output neuron, which can be written as

$$T(\boldsymbol{x};\boldsymbol{\theta}) = \boldsymbol{x}^T \prod_{i=1}^{L-1} \boldsymbol{W}_i \mathrm{dg}(\tilde{\boldsymbol{h}}_i) \boldsymbol{\alpha} \qquad (3)$$

where $\boldsymbol{\alpha}$ is a vector, $\mathrm{dg}(\tilde{\boldsymbol{h}}_i)$ is the diagonal matrix whose diagonal is the estimated value of $H_i$, and the meaning of rest symbols is the same as those of MLPCGE in definition 4. *Notice that both $\boldsymbol{x}$ and $\boldsymbol{h}_i$ are realizations or estimation of r.e.s, thus the Hessian of $T(\boldsymbol{x};\boldsymbol{\theta})$ is a random matrix* (Tao (2012)), which implies the Hessian $\boldsymbol{H}$ of $R(T)$ is a random matrix created by summing random matrices, each of which is a gradient $l'$ multiplies the Hessian of $T$.

*Thus, the problem converts to study the eigen-spectrum of a random matrix $\boldsymbol{H}$.* The conversion looks straightforward now, but is actually a major obstacle that stops Choromanska et al. (2015a) Jeffrey Pennington (2017), where a confusion about the source of randomness led them astray (the point is discussed in detail in appendix H.5). With the following *realistic* assumptions on $\boldsymbol{H}$ (for a discussion on the *practicality* of the assumptions, refer to appendix F.3), we show that $l$ has a surprising benign landscape. Suppose $\boldsymbol{H}$ is a $N \times N$ matrix, and let

$$\boldsymbol{A} := \mathbb{E}[\boldsymbol{H}], \frac{1}{\sqrt{N}} \boldsymbol{W} := \boldsymbol{H} - \boldsymbol{A}, \mathcal{S}[\boldsymbol{R}] := \frac{1}{N} \mathbb{E}_{\boldsymbol{W}}[\boldsymbol{W} \boldsymbol{R} \boldsymbol{W}]$$

where the expectation in $\mathcal{S}$ is taken w.r.t. $\boldsymbol{W}$ while keeping $\boldsymbol{R}$ fixed — it is a linear operator on the space of matrices.

**Assumption 5.1** (Boundedness). *1) $\exists C \in \mathbb{R}, \forall N \in \mathbb{N}, ||\boldsymbol{A}|| \leq C$, where $||\boldsymbol{A}||$ denotes the operator norm. 2) $\exists \mu_q \in \mathbb{R}, \forall q \in \mathbb{N}, \forall \alpha \in \mathbb{J}, \mathbb{E}[|\boldsymbol{W}_\alpha|^q] \leq \mu_q$, where $\mathbb{J} = \mathbb{I} \times \mathbb{I}$, and $\mathbb{I} = \{1, \ldots, N\}$. 3) $\exists C_1, C_2 \in \mathbb{R}, \forall R \in \mathbb{N}, \epsilon > 0, |||\kappa|||_2^{iso} \leq C_1, |||\kappa||| \leq C_2 N^\epsilon$; 4) $\exists 0 < c < C, \forall \boldsymbol{T} \succ \boldsymbol{0}, c\, N^{-1} tr\, \boldsymbol{T} \preceq \mathcal{S}[\boldsymbol{T}] \preceq C N^{-1} tr\, \boldsymbol{T}$.*

**Assumption 5.2** (Diversity). *There exists $\mu > 0$ such that the following holds: for every $\alpha \in \mathbb{I}$ and $q, R \in \mathbb{N}$, there exists a sequence of nested sets $\mathcal{N}_k = \mathcal{N}_k(\alpha)$ such that $\alpha \in \mathcal{N}_1 \subset \mathcal{N}_2 \subset \cdots \subset \mathcal{N}_R = \mathcal{N} \subset \mathbb{I}, |\mathcal{N}| \leq N^{1/2-\mu}$ and $\kappa(f(\boldsymbol{W}_{\mathbb{I} \setminus \cup_j \mathcal{N}_{n_j+1}(\alpha_j)}), g_1(\boldsymbol{W}_{\mathcal{N}_{n_1}(\alpha_1) \setminus \cup_{j \neq 1} \mathcal{N}(\alpha_j)}), \ldots, g_q(\boldsymbol{W}_{\mathcal{N}_{n_q}(\alpha_q) \setminus \cup_{j \neq q} \mathcal{N}(\alpha_j)})) \leq N^{-3q} ||f||_{q+1} \prod_{j=1}^q ||g_j||_{q+1}$, for any $n_1, \ldots, n_q < R$, $\alpha_1, \ldots, \alpha_q \in \mathbb{I}$ and real analytic functions $f, g_1, \ldots, g_q$, where $||||_p$ is the $L^p$ norm on function space. We call the set $\mathcal{N}$ of $\alpha$ the **coupling set** of $\alpha$.*

**Theorem 5.1.** *Let $R(T)$ be the risk function defined at eq. (1), where the loss function $l$ is of class $\mathcal{L}_0$, and the transport map $T$ is a neural network defined at eq. (3). If the Hessian $\boldsymbol{H}$ of $R(T)$ satisfies assumptions 5.1 5.2, $\mathbb{E}(\boldsymbol{H}) = \boldsymbol{0}$, and $N \to \infty$, then*

1. *all local minima are global minima with zero risk*
2. *A constant $\lambda_0 \in \mathbb{R}$ exists, such that the operator norm $||\boldsymbol{H}||$ of $\boldsymbol{H}$ is upper bounded by $\mathbb{E}_m[l'(T(X), Y)]\lambda_0$, where $\mathbb{E}_m[l'(T(X), Y)]$ is the empirical expectation of $l'$. It implies the regions around the minima are flat basins, where the eigen-spectrum of $\boldsymbol{H}$ is increasingly concentrated around zero.*

For the elaboration of the theorem, refer to appendix F. For the proof directly and further discussion of the theorem, refer to F.4. The error for finite $N$ is discussed at the remarks of theorem F.1.

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

## APPENDICES

All definitions present in the appendices are adopted and reproduced from existing literature with sources cited, for the purpose of making exact the terminology used in the paper.

## A  COUPLING THEORY

The following definitions are adapted from Thorisson (2000) unless otherwise noted.

**Definition A.1** (Random Element; Random Variable; Random Function). *A random element in a measurable space $(E, \mathcal{E})$ defined on a probability space $(F, \mathcal{F}, \mu)$ is a measurable mapping $T$ from $(F, \mathcal{F}, \mu)$ to $(E, \mathcal{E})$, where*

$$T^{-1}A \in \mathcal{F}, A \in \mathcal{E}; T^{-1}A := \{w \in F : T(w) \in A\}$$

*We say $T$ is* **supported by** *probability measure space $(F, \mathcal{F}, \mu)$, $(F, \mathcal{F}, \mu)$ is the support of $T$, $T$ is an $\mathcal{F}/\mathcal{E}$* **measurable mapping** *from $F$ to $E$, and the induced measure $\mu(T^{-1}(A))$ is the* **law** *of $T$. Some r.e.s have special names. When $(E, \mathcal{E})$ is the measurable space $(\mathbb{R}, \mathcal{B}(\mathbb{R}))$, where $\mathbb{R}$ is the real number field and $\mathcal{B}(\mathbb{R})$ is the Borel set, $X$ is also named as a* **random variable**, *whose abbreviation is r.v.; when $(E, \mathcal{E})$ is the multivariate real measurable space, $T$ is named as a multivariate r.v.; when $(E, \mathcal{E})$ is a function space satisfying certain conditions (Ionescu–Tulcea theorem, or Kolmogorov's extension theorem (Klenke (2012))), $T$ is named as a* **random function**.

**Definition A.2** (Coupling). *A probability measure $\mu$ on $\bigotimes_{i \in \mathbb{I}}(E_i, \mathcal{E}_i)$ is a coupling of $\mu_i, i \in \mathbb{I}$, if $\mu_i$ is the $i$th marginal of $\mu$, that is, if $\mu_i$ is induced by the $i$th projection mapping:*

$$\mu(\{x : x_i \in A\}) = \mu_i(A), A \in \mathcal{E}_i, i \in \mathbb{I}$$

*where $\mathbb{I}$ is an index set, $\mu_i$ is a probability measure on a measurable space $(E_i, \mathcal{E}_i)$, and $\bigotimes_{i \in \mathbb{I}}(E_i, \mathcal{E}_i) := (\prod_{i \in \mathbb{I}} E_i, \bigotimes_{i \in \mathbb{I}} \mathcal{E}_i)$, $\prod_{i \in \mathbb{I}} E_i$ is the Cartesian product of $E_i$ and $\bigotimes_{i \in \mathbb{I}} \mathcal{E}_i$ is the product $\sigma$-algebra.*

The general idea of coupling is to find a dependence structure (joint distribution) from fixed marginal distributions that fits one's purpose.

**Definition A.3** (Coupling of Random Element; Deterministic Coupling; Transport Map). *Given two r.e.s $X, Y$, a coupling $(X, Y)$ refers to the coupling of probability measure $\mu, \nu$ of probability space $(E, \mathcal{E}, \mu), (F, \mathcal{F}, \nu)$, where $\mu, \nu$ is the probability measure of r.e. $X, Y$ respectively. The coupling $(X, Y)$ is called* **deterministic** *if there exists a measurable function $T : \mathcal{E} \to \mathcal{F}$ such that $Y = T(X)$. $T$ is normally referred as* **transport map**. *Informally, we say that $T$ transports measure $\mu$ of $X$ to measure $\nu$ of $Y$. The definitions are adapted from Villani (2008).*

To recognize an event $\omega_s$ that is composed of events of the lowest detectable scale $\Omega_{s_0}$, the idea is to transport the probability measure of the event $\omega_s$ through a deterministic coupling, and construct a r.e. that represents possible states $\omega_s$ may take. We introduce concepts needed in the following.

**Definition A.4** (Extension of Probability Space). *A probability space $(\bar{F}, \bar{\mathcal{F}}, \bar{\mu})$ is an extension of another probability space $(F, \mathcal{F}, \mu)$, if $(\bar{F}, \bar{\mathcal{F}}, \bar{\mu})$ supports a r.e. $\xi$ in $(F, \mathcal{F})$ having law $\mu$. If $T$ is a r.e. in $(E, \mathcal{E})$ defined on $(F, \mathcal{F}, \mu)$, then it has a* **copy** *$\bar{T}$, i.e., the r.e. $\bar{T}$ defined on $(\bar{F}, \bar{\mathcal{F}}, \bar{\mu})$ by $\bar{T}(\bar{\omega}) = T(\xi(\bar{\omega})), \bar{\omega} \in \bar{\Omega}$ and $\bar{\mu}(\bar{T}(\cdot)) = \mu(T(\cdot))$. $\bar{T}$ is called* **original** *r.e.. New r.e.s may be created, which is called* **external** *r.e.s.*

The goal of S-System is to create extensions of a measurement collection r.e. $\mathcal{X}$ to reconstruct $\Omega_s$ for some $s \in \mathcal{S}$ in PPMS, such that $\mu^{\mathcal{X}}(H_s = h_s) = \mu^{\mathcal{W}}(\omega_s)$, where $H_s$ is a r.e. created through extension, and $h_s$ is a realized value of it. However, we do not possess the p.d.f. of $X$, which we have to rely on realizations of $X$. What we can do is to leverage and only leverage available information through conditioning extension.

**Definition A.5** (Probability Kernel). *Given two measurable space $(E_1, \mathcal{E}_1), (E_2, \mathcal{E}_2)$, a function $Q(\cdot, \cdot)$ from $E_1 \times \mathcal{E}_2$ to $[0, 1]$ is an $((E_1, \mathcal{E}_1), (E_2, \mathcal{E}_2))$ probability kernel if 1) $Q(\cdot, A)$ is $\mathcal{E}_1/\mathcal{B}([0, 1])$ measurable for each $A \in \mathcal{E}_2$ 2) and $Q(y, \cdot)$ is probability measure on $(E_2, \mathcal{E}_2)$ for each $y \in E_1$. A probability kernel uniquely determines a probability measure on $(E_1, \mathcal{E}_1) \bigotimes (E_2, \mathcal{E}_2)$ (Ash (1972) Section 2.6.2).*

**Definition A.6** (Conditioning Extension). *Let $T_1$ be an r.e. in $(E_1, \mathcal{E}_1)$ defined on a probability measure space $(F, \mathcal{F}, \mu)$, and let $Q(\cdot, \cdot)$ be an $((E_1, \mathcal{E}_1), (E_2, \mathcal{E}_2))$ probability kernel. A conditioning extension $(\bar{F}, \bar{\mathcal{F}}, \bar{\mu})$ of $(F, \mathcal{F}, \mu)$ is created by letting*

$$(\bar{F}, \bar{\mathcal{F}}) := (F, \mathcal{F}) \bigotimes (E_2, \mathcal{E}_2) \qquad \bar{\mu}(A \times B) := \int_A Q(T(\omega), B) \mu(d\omega), A \in \mathcal{F}, B \in \mathcal{E}_2$$

$$\xi(\omega, t) := \omega, \omega \in \Omega, t \in E_2 \qquad \bar{T}_1(\omega, t) := T_1(\omega), \omega \in \Omega, t \in E_2 \quad \bar{T}_2(\omega, t) := t, t \in E_2$$

*$\bar{T}_1$ is the original r.e.s in the new probability space, while $\bar{T}_2$ is a new external r.e. created. Conditioning extension can be repeated countably many times (Ash (1972) Section 2.7.2).*

**Definition A.7** (Splitting Extension). *Let $T_1, T_2$ be r.e.s in $(E_1, \mathcal{E}_1), (E_2, \mathcal{E}_2)$ respectively, defined on a probability space $(F, \mathcal{F}, \mu)$. Let $\nu$ be a probability measure on a Polish space $(E_3, \mathcal{E}_3)$, let $Q(\cdot, \cdot)$ be an $((E_3, \mathcal{E}_3), (E_2, \mathcal{E}_2))$ probability kernel, and suppose*

$$\mu(T_2 \in A) = \int Q(t, A) \nu(dt), A \in \mathcal{E}_2$$

*Then a splitting extension of $(F, \mathcal{F}, \mu)$ is to create an extension to support a r.e. $T_3$ in $(E_3, \mathcal{E}_3)$ having distribution $\nu$, and such that*

$$\mu(T_2 \in \cdot | T_3 = t) = Q(t, \cdot), t \in \mathcal{E}_3$$

*Furthermore, $T_1$ is conditionally independent of $T_3$ given $T_2$.*

## B  S-SYSTEM DETAILS

---
**Algorithm 2** Example implementations of S-System functions `successor` and `predecessor`.

---
**function** SUCCESSOR($s$)
    **return** the set of elements in $\mathcal{S}^{\mathcal{X}}$ that are the immediate successors of $s$ (immediate successors of $s$ is the set of smallest elements that are comparable with $s$ and larger than $s$, though themselves are not comparable)
**end function**
**function** PREDECESSOR($s$)
    **return** the set of indices of elements in $\mathcal{S}^{\mathcal{X}}$ that are the immediate predecessors of $s$ (immediate predecessors of $s$ is the set of smallest elements that are comparable with $s$ and smaller than $s$, though themselves are not comparable)
**end function**

---

## C  MANIFOLD AND DIFFEOMORPHISM GROUP

The following definitions have been adapted from Lee (2012) unless otherwise noted.

**Definition C.1** (Topological Manifold). *Suppose $M$ is a topological space. We say that $M$ is a* **topological manifold of dimension n** *or a* **topological n-manifold** *or just a* **n-manifold** *if it has the following properties:*

- *$M$ is a* **Hausdorff space***: for every pair of distinct points $p, q \in M$, there are disjoint open subsets $U, V \subseteq M$ such that $p \in U$ and $q \in V$.*
- *$M$ is* **second-countable***: there exists a countable basis for the topology of $M$.*
- *$M$ is* **locally Euclidean of dimension n***: each point of $M$ has a neighborhood that is homeomorphic to an open subset of $\mathbb{R}^n$.*

**Definition C.2** (Smooth Mapping; Diffeomorphism). *If $U, V$ are open subsets of Euclidean spaces $\mathbb{R}^n$ and $\mathbb{R}^m$, respectively, a function $f : U \rightarrow V$ is said to be* **smooth** *(or $C^\infty$, or* **infinitely differentiable***) if each of its component functions has continuous partial derivatives of all orders. If in addition $f$ is bijective and has a smooth inverse map, it is called a* **diffeomorphism***.*

**Definition C.3** (Chart; Coordinate Chart; Smooth Compatible). *Let $M$ be a topological $n$-manifold. A* **coordinate chart** *(or just a* **chart***) on $M$ is a pair $(U, \psi)$, where $U$ is an open subset of $M$ and*

$\psi : U \to \hat{U}$ is a homeomorphism from $U$ to an open subset $\hat{U} = \psi(U) \subseteq \mathbb{R}^n$. $U$ is called a **coordinate domain**, or just **domain**, $\psi$ is called a **(local) coordinate map**, and the component functions $(x^1, \ldots, x^n)$ of $\psi$, defined by $p \in M, \psi(p) = (x^1(p), \ldots, x^n(p))$, are called **local coordinates** on $U$. Two $(U, \phi), (V, \psi)$ are called **smoothly compatible** if either $U \cap V = \emptyset$, or $\psi \circ \phi^{-1}$ is a diffeomorphism.

**Definition C.4** (Atlas; Smooth Atlas; Maximal Atlas). *Let $M$ be a topological manifold. An **atlas** $\mathcal{A}$ for $M$ is a collection of charts whose domain cover $M$. If any two charts in $\mathcal{A}$ is smoothly compatible with each other, it is called a **smooth atlas**. A smooth atlas $\mathcal{A}$ on $M$ is **maximal** if it is not properly contained in any larger smooth atlas.*

**Definition C.5** (Smooth Manifold). *A **smooth manifold** is a pair $(M, \mathcal{A})$, where $M$ is a topological manifold and $\mathcal{A}$ is a maximal smooth atlas on $M$. When no confusion exists, we may just say "$M$ is a smooth manifold".*

**Definition C.6** (Riemannian Metric). *A **Riemannian metric** $g$ of a smooth manifold $M$ is a symmetric covariant 2-tensor field on $M$ that is positive definite at each point. It defines an inner product on $M$, which informally, could be represented by a quadratic form $\boldsymbol{\eta}^T \boldsymbol{G} \boldsymbol{\eta}'$, where $\boldsymbol{G} = (g_{ij})$ is a matrix and $\boldsymbol{\eta}, \boldsymbol{\eta}'$ are the local coordinates of two points in $M$.*

**Definition C.7** (Riemannian Manifold). *A **Riemannian manifold** is a pair $(M, g)$, where $M$ is a smooth manifold and $g$ is a Riemannian manifold. Or in short, we could say $M$ is a Riemannian manifold if $M$ is understood to be endowed with a specific Riemannian metric.*

The following definition is adopted from Banyaga (1997).

**Definition C.8** ($C^\infty$ Diffeomorphism Group). *Let $C^\infty(M, N)$ denote the space of all $C^\infty$ mapping $f : M \to N$, where $M, N$ are smooth manifolds. The diffeomorphism group, denoted by $\text{Diff}^\infty(M)$, is the set of all $C^\infty$ diffeomorphisms of $M$, the group action of which is the mapping composition.*

To make the definition more concrete to help understanding, we provide an example adopted from Mallat (2016).

**Example C.1.** *The diffeomorphism group is the set of deformation that may be applied to objects, e.g., images, of which we can define a norm to characterize the deformation. A small diffeomorphism acting on a function $x(u)$ defined on $\mathbb{R}^n$ can be written as a translation of $u$ by a $g(u)$:*

$$g.x(u) = x(u - g(u)), g \in \text{Diff}^\infty(\mathbb{R}^n)$$

*Note that the smooth condition is not necessary, and is only used to avoid introducing further definitions. The diffeomorphism translates points by at most $||g||_\infty = \sup_{u \in \mathbb{R}^n} |g(u)|$. Small diffeomorphism corresponds to $||\nabla g||_\infty = \sup_u |\nabla g(u)| < 1$, where $|\nabla g|$ is the matrix norm of the Jacobian matrix of $g$ at $u$. Thus, an element in a subset of $\text{Diff}^\infty(\mathbb{R}^n)$ can be understood as a small deformation of images where the deformation is bounded.*

## D  INFORMATION GEOMETRY

The following definitions are adapted from Amari (2016).

**Definition D.1** (Divergence). *Suppose that $M$ is a $n$-manifold, of which the points have a local coordinates system $\eta$. Given two points $\boldsymbol{p}, \boldsymbol{q} \in M$, the coordinates of which are $\boldsymbol{\eta}_p, \boldsymbol{\eta}_q$ respectively, a **divergence** is a function of $\boldsymbol{\eta}_p, \boldsymbol{\eta}_q$, written as $D[\boldsymbol{p} : \boldsymbol{q}]$ or $D[\boldsymbol{\eta}_p : \boldsymbol{\eta}_q]$, which satisfies the following criteria.*

- *$D[\boldsymbol{p} : \boldsymbol{q}] \geq 0$.*
- *$D[\boldsymbol{p} : \boldsymbol{q}] = 0$, if and only if $\boldsymbol{p} = \boldsymbol{q}$.*

*When $\boldsymbol{p}$ and $\boldsymbol{q}$ are sufficiently close, and $D$ is differentiable, by denoting their coordinates by $\boldsymbol{\eta}_p$ and $\boldsymbol{\eta}_q = \boldsymbol{\eta}_p + d\boldsymbol{\eta}$, the Taylor expansion of $D$ is written as*

$$D[\boldsymbol{\eta}_p : \boldsymbol{\eta}_q + d\boldsymbol{\eta}] = \frac{1}{2} \sum g_{ij}(\boldsymbol{\eta}_p) d\boldsymbol{\eta}_i \boldsymbol{\eta}_j + O(|d\boldsymbol{\eta}|^3)$$

*, and matrix $\boldsymbol{G} = (g_{ij})$ is positive-definite, depending on $\boldsymbol{\eta}_p$.*

**Definition D.2** (Bregman Divergence). *Given a convex function $\psi(\boldsymbol{\eta})$, a **Bregman divergence** derived from $\psi$ is a divergence defined as*

$$D_\psi[\boldsymbol{\eta} : \boldsymbol{\eta}'] = \psi(\boldsymbol{\eta}') - \psi(\boldsymbol{\eta}) - \nabla\psi(\boldsymbol{\eta})^T(\boldsymbol{\eta}' - \boldsymbol{\eta})$$

**Definition D.3** (Legendre Dual; Legendre Transform). *Given a convex function $\psi(\boldsymbol{\eta})$, the **Legendre dual** of $\psi$ is the function $\psi^*$*

$$\psi^*(\boldsymbol{\eta}^*) = \boldsymbol{\eta}^T \boldsymbol{\eta}^* - \psi(\boldsymbol{\eta})$$

*where $\boldsymbol{\eta} = \boldsymbol{f}(\boldsymbol{\eta}^*)$ and $\boldsymbol{f}$ is the inverse function of $\boldsymbol{\eta}^* = \nabla\psi(\boldsymbol{\eta})$. $\psi^*$ is a convex function. $\nabla\psi(\boldsymbol{\eta})$ is called the **Legendre Transform** of $\eta$.*

**Definition D.4** (Dual Bregman Divergence). *Given a convex function $\psi(\boldsymbol{\eta})$, and $D_\psi$ the Bregman divergence derived by $\psi$. Let $\psi^*$ be the Legendre dual function of $\psi$, then the Bregman divergence $D_{\psi^*}$ defined by $\psi^*$ is called the **Dual Bregman Divergence** derived by $\psi$. We have*

$$D_{\psi^*}[\boldsymbol{\eta}^* : \boldsymbol{\eta}'^*] = D_\psi[\boldsymbol{\eta}' : \boldsymbol{\eta}]$$

**Definition D.5** (Exponential Family of Probability Distributions; Stochastic Manifold; Affine Coordinate System; Dual Affine Coordinate System). *The form of **probability distribution of exponential family** is given by the probability density function*

$$p(\boldsymbol{x}; \boldsymbol{\theta})d\boldsymbol{x} = e^{(\boldsymbol{\theta}^T \boldsymbol{h}(\boldsymbol{x}) - \psi(\boldsymbol{\theta}))} d\mu(\boldsymbol{x})$$

*where $\boldsymbol{x}$ is a realizable value of a multivariate random variable, $\boldsymbol{h}(\boldsymbol{x})$ is a vector function of $\boldsymbol{x}$ which are linearly independent, $\psi(\boldsymbol{\theta})$ is the normalization factor, and $\mu$ is the law on r.v. $\boldsymbol{x}$.*

*Since $\psi$ is a convex function w.r.t. $\boldsymbol{\theta}$, the exponential family distributions is a Riemannian manifold $(M, g)$ with an **affine coordinate system** $\boldsymbol{\theta} = (\theta_1, \ldots, \theta_n)$, and $g$ is given by the second order Taylor expansion of the Bregman divergence derived from $\psi$. It is called the **stochastic manifold of exponential family distribution**. $\boldsymbol{\theta}$ is called natural or canonical parameters. An alternative coordinate system of $M$ is given by the Legendre transform*

$$\boldsymbol{\eta} = \nabla\psi(\boldsymbol{\theta}) = \int \boldsymbol{h} e^{\boldsymbol{\theta}^T \boldsymbol{h}(\boldsymbol{x})} d\mu(\boldsymbol{x})$$

*of $\boldsymbol{\theta}$, which is well known as the expectation parameter in statistics, and is called **dual affine coordinate system**. Correspondingly, an alternative Riemannian metric is derived from the Legendre dual of $\psi$. The Bregman divergence derived is the well known discrepancy measure on probability, the KL divergence.*

# E  STATISTICAL LEARNING THEORY

Assume a sample space $\mathcal{Z} = \mathcal{X} \times \mathcal{Y}$, where $\mathcal{X}$ is the space of input data, and $\mathcal{Y}$ is the label space. We use $S_m = \{s_i = (\boldsymbol{x}_i, y_i)\}_{i=1}^m$ to denote the training set of size $m$ whose samples are drawn independently and identically distributed (i.i.d.) according to an unknown distribution $P$. Given a loss function $l$, the goal of learning is to identify a function $f : \mathcal{X} \mapsto \mathcal{Y}$ in a hypothesis space (a class $\mathcal{F}$ of functions) that minimizes the expected error

$$R(f) = \mathbb{E}_{z \sim P}\left[l\left(f(\boldsymbol{x}), y\right)\right],$$

where $z = (\boldsymbol{x}, y) \in \mathcal{Z}$ is sampled i.i.d. according to $P$. Since $P$ is unknown, the observable quantity serving as the proxy to the true risk $R(f)$ is the empirical error

$$R_m(f) = \frac{1}{m} \sum_{i=1}^m l\left(f(\boldsymbol{x}_i), y_i\right).$$

# F  DERIVATION, PROOF, AND FURTHER INTERPRETATION OF THEOREM 5.1

## F.1  HESSIAN OF NN IS INHERENTLY A HUGE RANDOM MATRIX

As explained in section 5, to study the landscape of the loss function, we study the eigenvalue distribution of its Hessian $\boldsymbol{H}$ at the critical points. First, we derive the Hessian $\boldsymbol{H}$ of loss function of class $\mathcal{L}_0$ composed upon NNs with a single output. For a review of matrix calculus, the reader may refer to Magnus & Neudecker (2007).

The first partial differential of $l$ w.r.t. $\boldsymbol{W}_p$ is

$$\partial l(T\boldsymbol{x}, y) = l'(T\boldsymbol{x}, y)\boldsymbol{\alpha}^T \prod_{j=p+1}^{L-1} (\mathrm{dg}(\tilde{\boldsymbol{h}}_j')\boldsymbol{W}_j^T)\mathrm{dg}(\tilde{\boldsymbol{h}}_p') \otimes \boldsymbol{x}^T \prod_{i=1}^{p-1}(\boldsymbol{W}_i\mathrm{dg}(\tilde{\boldsymbol{h}}_i))\partial\mathrm{vec}\boldsymbol{W}_p$$

where $\otimes$ denotes Kronecker product. Note that for clarity of presentation, we use the partial differential the same way as differential is defined and used in Magnus & Neudecker (2007), i.e., $\partial l$ is a number instead of an infinitely small quantities, though in the book, partial differential is not defined explicitly. $\tilde{h}'$ is an abuse of notation for clarity and needs some explanation. Recall that $\{h_i\}_{i=1,\dots,L-1}$ are the group indicator r.e.s, and $\{\tilde{h}_i\}_{i=1,\dots,L-1}$ are the estimation of them based on transport maps. $\tilde{h}_i$ is a scalar function, and denote it as $\tilde{h}_i(a)$, where $a$ is the computed input. When computing the partial differential w.r.t. $W_p$, by the chain rule, the differential of $\tilde{h}_i(a)$ w.r.t. $W_p$ is $\partial\tilde{h}_i(a)/\partial a$. To avoid introducing too much clutter, we denote $\tilde{h}_i'$ as $\partial\tilde{h}_i(a)/\partial a$.

Since $H$ is symmetric, we only need to compute the block matrices by taking partial differential w.r.t. $W_q$, where $q > p$ — taking partial differential w.r.t. $W_p$ again gives zero matrix.

$$
\begin{aligned}
\partial^2 l(T\boldsymbol{x}, y) =&\, l'(T\boldsymbol{x}, y) \\
&[(\boldsymbol{\alpha}^T \prod_{k=q+1}^{L-1} (\mathrm{dg}(\tilde{\boldsymbol{h}}_k'')\boldsymbol{W}_k^T)\mathrm{dg}(\tilde{\boldsymbol{h}}_q'') \otimes \mathrm{dg}(\tilde{\boldsymbol{h}}_p') \prod_{j=p+1}^{q-1} (\boldsymbol{W}_j\mathrm{dg}(\tilde{\boldsymbol{h}}_j'))\partial\mathrm{vec}\boldsymbol{W}_q)^T \\
&\otimes \boldsymbol{x}^T \prod_{i=1}^{p-1} (\boldsymbol{W}_i\mathrm{dg}(\tilde{\boldsymbol{h}}_i))]\partial\mathrm{vec}\boldsymbol{W}_p \\
=&\, l'(T\boldsymbol{x}, y) \\
&(\partial\mathrm{vec}\boldsymbol{W}_q)^T[\mathrm{dg}(\tilde{\boldsymbol{h}}_q'') \prod_{k=q+1}^{L-1} (\boldsymbol{W}_k\mathrm{dg}(\tilde{\boldsymbol{h}}_k''))\boldsymbol{\alpha} \otimes \prod_{j=p+1}^{q-1} (\mathrm{dg}(\tilde{\boldsymbol{h}}_j')\boldsymbol{W}_j^T)\mathrm{dg}(\tilde{\boldsymbol{h}}_p') \\
&\otimes \boldsymbol{x}^T \prod_{i=1}^{p-1} (\boldsymbol{W}_i\mathrm{dg}(\tilde{\boldsymbol{h}}_i))]\partial\mathrm{vec}\boldsymbol{W}_p
\end{aligned}
$$

where again $\tilde{h}''$ is an abuse of notation, it is actually $\tilde{h}' \odot \tilde{h}'$, the hamadard product of partial differentials obtained by taking partial differential w.r.t. $W_p$, and w.r.t. $W_q$. The two partial differentials are the same because $a$, the input the $h$, is the same throughout. Notice that the estimation $\tilde{h}$ of group indicator r.e.s $H$ is merely a function of $\hat{H}$. It implies $\tilde{h}$ is a realization of a r.e. created by a deterministic coupling by applying a transport map on $\hat{H}$. The deeper principles of the estimation will be explained in appendix G. For now, it suffices to stop with the fact that the entries of $H$ is a random variable. As an example, for estimation done by ReLU, $\tilde{h}$ would be $\tilde{h}_i = \max\{0, \hat{h}_i\}$. Thus, $H$ is an ensemble of real symmetric random matrix with correlated entries. Denote

$$
\boldsymbol{H}_{pq} = l'(T\boldsymbol{x}, y)\mathrm{dg}(\tilde{\boldsymbol{h}}_q'') \prod_{k=q+1}^{L-1} (\boldsymbol{W}_k\mathrm{dg}(\tilde{\boldsymbol{h}}_k''))\boldsymbol{\alpha} \otimes \prod_{j=p+1}^{q-1} (\mathrm{dg}(\tilde{\boldsymbol{h}}_j')\boldsymbol{W}_j^T)\mathrm{dg}(\tilde{\boldsymbol{h}}_p') \otimes \boldsymbol{x}^T \prod_{i=1}^{p-1} (\boldsymbol{W}_i\mathrm{dg}(\tilde{\boldsymbol{h}}_i))
$$

(4)

We have the Hessian of $l$ as

$$
\boldsymbol{H} = \begin{bmatrix} \boldsymbol{0} & \boldsymbol{H}_{12}^T & \dots & \boldsymbol{H}_{1L}^T \\ \boldsymbol{H}_{12} & \boldsymbol{0} & \dots & \boldsymbol{H}_{2L}^T \\ \vdots & \ddots & \ddots & \vdots \\ \boldsymbol{H}_{1L} & \boldsymbol{H}_{2L} & \dots & \boldsymbol{0} \end{bmatrix}
$$

(5)

## F.2 EIGENVALUE DISTRIBUTION OF SYMMETRY RANDOM MATRIX WITH SLOW CORRELATION DECAY

In this section, we show how to obtain the eigen-spectrum of $H$ through random matrix theory (RMT) (Tao (2012)). RMT has been born out of the study on the nuclei of heavy atoms, where the spacings between lines in the spectrum of a heavy atom nucleus is postulated the same with spacings between eigenvalues of a random matrix (Wigner (1957)). In a certain way, it seems to be the backbone math of complex systems, where the collective behaviors of sophisticated subunits can be analyzed stochastically when deterministic or analytic analysis is intractable.

The following definitions can be found in Tao (2012) unless otherwise noted.

The eigen-spectrum is studied as empirical spectral distribution (ESD) in RMT, define as

**Definition F.1** (Empirical Spectral Distribution). *Given a $N \times N$ random matrix $\boldsymbol{H}$, its **empirical spectral distribution** $\mu_{\boldsymbol{H}}$ is*

$$\mu_{\boldsymbol{H}} = \frac{1}{N} \sum_{i=1}^{N} \delta_{\lambda_i}$$

*where $\{\lambda_i\}_{i=1,\dots,N}$ are all eigenvalues of $\boldsymbol{H}$ and $\delta$ is the delta function.*

Given a hermitian matrix $\boldsymbol{H}$, its ESD $\mu_{\boldsymbol{H}}(\lambda)$ can be studied via its resolvent $\boldsymbol{G}$.

**Definition F.2** (Resolvent). *Let $\boldsymbol{H}$ be a normal matrix, and $z \in \mathbb{H}$ a spectral parameter. The **resolvent** $\boldsymbol{G}$ of $\boldsymbol{H}$ at $z$ is defined as*

$$\boldsymbol{G} = \boldsymbol{G}(z) = \frac{1}{\boldsymbol{H} - z}$$

*where*

$$\mathbb{H} := \{z \in \mathbb{C} : \Im z > 0\}$$

$\mathbb{C}$ *denotes the complex field, and $\Im$ is the function gets imaginary part of a complex number $z$.*

$\boldsymbol{G}$ compactly summarizes the spectral information of $\boldsymbol{H}$ around $z$, which is normally analyzed by *functional calculus* on operators, and defined through Cauchy integral formula on operators

**Definition F.3** (Functions of Operators).

$$f(T) = \frac{1}{2\pi i} \int_C \frac{f(\lambda)}{\lambda - T} d\lambda \tag{6}$$

*where $f$ is an analytic scalar function and $C$ is an appropriately chosen contour in $\mathbb{C}$.*

The formula can be defined on a range of linear operators (Dunford & Schwartz (1957)) (Recall that a linear operator is a mapping whose domain and codomain are defined on the same field). Since the most complex case involved here will be a normal matrix, we stop at stating that the formula holds true when $T$ is a normal matrix.

Resolvent $\boldsymbol{G}$ is related to eigen-spectrum of $\boldsymbol{H}$ through stieltjes transform of $\mu_{\boldsymbol{H}}(\lambda)$.

**Definition F.4** (Stieltjes Transform). *Let $\mu$ be a Borel probability measure on $\mathbb{R}$. Its **Stieltjes transform** at a spectral parameter $z \in \mathbb{H}$ is defined as*

$$m_\mu(z) = \int_{\mathbb{R}} \frac{d\mu(x)}{x - z}$$

With some efforts, it can be seen that the normalized trace of $\boldsymbol{G}$ is stieltjes transform of eigen-spectrum of $\boldsymbol{H}$

$$m_{\mu_{\boldsymbol{H}}}(z) = \frac{1}{N} \mathrm{tr}\, G$$

For a proof, the reader may refer to proposition 2.1 in Alt et al. (2018). $\mu_{\boldsymbol{H}}$ can be recovered from $m_{\mu_{\boldsymbol{H}}}$ through the inverse formula of Stieltjes-Perron.

**Lemma F.1** (Inverse Stieljies Transform). *Suppose that $\mu$ is a probability measure on $\mathbb{R}$ and let $m_\mu$ be its Stieltjes transform. Then for any $a < b$, we have*

$$\mu((a, b)) + \frac{1}{2}[\mu(\{a\}) + \mu(\{b\})] = \lim_{\Im z \to 0} \frac{1}{\pi} \int_b^a \Im m_\mu(z) d\Re z$$

*where $\Re$ is the function that gets the real part of $z$. The proof can be found at Tao (2012) p. 144.*

Consequently, the problem converts to obtain $\boldsymbol{G}$ if we want to obtain $\mu_{\boldsymbol{H}}$. A recent advance in the RMT community has enabled the analysis of ESD of symmetric random matrix with correlation (Erdős et al. (2017)) from the perspective of mean field theory (Kadanoff (2000)), which we debrief in the following.

The resolvent $\boldsymbol{G}$ holds an identity by definition

$$\boldsymbol{H}\boldsymbol{G} = \boldsymbol{I} + z\boldsymbol{G} \tag{7}$$

Note that in the above equation $\boldsymbol{G}$ is a function $\boldsymbol{G}(\boldsymbol{H})$ of $\boldsymbol{H}$. When the average fluctuation of entries of $\boldsymbol{G}$ w.r.t. to its mean is small as $N$ grows large, eq. (7) can be turned into a solvable equation regarding $\boldsymbol{G}$ instead of merely a definition. Formally, it is achieved by taking the expectation of eq. (7)

$$\mathbb{E}[\boldsymbol{H}\boldsymbol{G}] = \boldsymbol{I} + z\mathbb{E}[\boldsymbol{G}] \tag{8}$$

When fluctuation of moments beyond the second order are negligible, we can obtain a class of random matrices whose ESD can be obtained by solving a truncated cumulant expansion of eq. (8). With the above approach, using sophisticated multivariate cumulant expansion, Erdős et al. (2017) proves $\boldsymbol{G}$ can be obtained as the unique solution to *Matrix Dyson Equation (MDE)* below

$$\boldsymbol{I} + (z - \boldsymbol{A} + \mathcal{S}[\boldsymbol{G}])\boldsymbol{G} = \boldsymbol{0}, \Im\boldsymbol{G} \succ \boldsymbol{0}, \Im z > 0 \tag{9}$$

where $\Im\boldsymbol{G} \succ 0$ means $\Im\boldsymbol{G}$ is positive definite,

$$\boldsymbol{A} := \mathbb{E}[\boldsymbol{H}], \frac{1}{\sqrt{N}}\boldsymbol{W} := \boldsymbol{H} - \boldsymbol{A}, \mathcal{S}[\boldsymbol{R}] := \frac{1}{N}\mathbb{E}_{\boldsymbol{W}}[\boldsymbol{W}\boldsymbol{R}\boldsymbol{W}] \tag{10}$$

$\mathcal{S}$ is a linear map on the space of $N \times N$ matrices and $\boldsymbol{W}$ is a random matrix with zero expectation. The expectation is taken w.r.t. $\boldsymbol{W}$, while taking $\boldsymbol{R}$ as a deterministic matrix.

We describe their results formally in the following, which begins with some more definitions, adopted from Erdős et al. (2017).

**Definition F.5** (Cumulant). **Cumulants** *of $\kappa_{\boldsymbol{m}}$ of a random vector $\boldsymbol{w} = (w_1, \dots, w_n)$ are defined as the coefficients of log-characteristic function*

$$\log \mathbb{E}e^{i\boldsymbol{t}^T\boldsymbol{w}} = \sum_{\boldsymbol{m}} \kappa_{\boldsymbol{m}} \frac{i\boldsymbol{t}^{\boldsymbol{m}}}{\boldsymbol{m}!}$$

*where $\sum_{\boldsymbol{m}}$ is the sum over all $n$-dimensional multi-indices $\boldsymbol{m} = (m_1, \dots, m_n)$.*

To recall, a multi-indices is

**Definition F.6** (Multi-index). *a $n$-dimensional multi-index is an $n$-tuple*

$$\boldsymbol{m} = (m_1, \dots, m_n)$$

*of non-negative integers. Note that $|\boldsymbol{m}| = \sum_{i=1}^n m_i$, and $\boldsymbol{m}! = \prod_{i=1}^n m_i!$.*

Similar with the more familiar concept *moment*, cumulant is also a measure of statistic properties of r.e.s. Particularly, the $k$-order cumulant $\kappa$ characterizes the $k$-way correlation of a set of r.v.s. The key of insight of the paper is to identify the condition where a matrix entry $w_\alpha, \alpha \in \mathbb{I}$ is only strongly correlated with a minority of $\boldsymbol{W}_{\mathbb{I}\backslash\{\alpha\}}$, and higher order cumulants tend to be weak and not influential in large $N$ limit. Thus, a proper formulation of the correlation strength is needed, and is defined as the cumulant norms on entries of $\boldsymbol{W}$ in the following. Given $k$ entries $\boldsymbol{W}_{\boldsymbol{\alpha}}$ at $\boldsymbol{\alpha} = \{\alpha_i\}_{i=1,\dots,k}, \alpha_i \in \mathbb{I}$ of matrix $\boldsymbol{W}$, where duplication is allowed, denote $\kappa(\alpha_1, \dots, \alpha_k) = \kappa(w_{\alpha_1}, \dots, w_{\alpha_k})$.

**Definition F.7** (Cumulant Norms).

$$|||\kappa||| := |||\kappa|||_{\leq R} := \max_{2 \leq k \leq R} |||\kappa|||_k, |||\kappa|||_k := |||\kappa|||_k^{av} + |||\kappa|||_k^{iso}$$

$$|||\kappa|||_2^{av} := || |\kappa(*,*)| ||, |||\kappa|||_k^{av} := N^{-2} \sum_{\alpha_1,\dots,\alpha_k} |\kappa(\alpha_1,\dots,\alpha_k)|, k \geq 4 \tag{11a}$$

$$|||\kappa|||_3^{av} := || \sum_{\alpha_1} |\kappa(\alpha_1,*,*)| || + \inf_{\kappa=\kappa_{dd}+\kappa_{dc}+\kappa_{cd}+\kappa_{cc}} (|||\kappa_{dd}|||_{dd} + |||\kappa_{dc}|||_{dc} + |||\kappa_{cd}|||_{cd} + |||\kappa_{cc}|||_{cc})$$

$$\tag{11b}$$

$$|||\kappa|||_{cc} = |||\kappa|||_{dd} := N^{-1} \sqrt{\sum_{b_2,a_3} (\sum_{a_2,b_3} \sum_{\alpha_1} |\kappa(\alpha_1, a_2b_2, a_3b_3)|)^2}$$

$$|||\kappa|||_{cd} := N^{-1} \sqrt{\sum_{b_3,a_1} (\sum_{a_3,b_1} \sum_{\alpha_2} |\kappa(a_1b_1, \alpha_2, a_3b_3)|)^2}, |||\kappa|||_{dc} := N^{-1} \sqrt{\sum_{b_1,a_2} (\sum_{a_1,b_2} \sum_{\alpha_3} |\kappa(a_1b_1, a_2b_2, \alpha_3)|)^2}$$

$$|||\kappa|||_2^{iso} := \inf_{\kappa=\kappa_d+\kappa_c} (|||\kappa_d|||_d + |||\kappa_c|||_c), |||\kappa_d|||_d := \sup_{||\boldsymbol{x}||\leq 1} |||| \kappa(\boldsymbol{x}*,\cdot*)| ||, |||\kappa|||_c := \sup_{||\boldsymbol{x}||\leq 1} || |\kappa(\boldsymbol{x}*,*\cdot)| ||$$

$$\tag{11c}$$

$$|||\kappa|||_k^{iso} := || \sum_{\alpha_1,\dots,\alpha_{k-2}} |\kappa(\alpha_1,\dots,\alpha_{k-2},*,*)| ||, k \geq 3$$

*where in eq. (11b), the infimum is taken over all decomposition of $\kappa$ in four symmetric functions $\kappa_{dd}, \kappa_{dc}, \kappa_{cd}, \kappa_{cc}$; in eq. (11c) the infimum is taken over all decomposition of $\kappa$ into the sum of symmetric $\kappa_c$ and $\kappa_d$. The norms defined in eq. (11a) and eq. (11c) need some explanation on the notation. If, in place of an index $\alpha \in \mathbb{J}$, we write a dot $(\cdot)$ in a scalar quantity then we consider the quantity as a vector indexed by the coordinate at the place of the dot. For example, $\kappa(a_1\cdot, a_2b_2)$ is a vector, the $i$-th entry of which is $\kappa(a_1i, a_2b_2)$ and therefore the inner norms in eq. (11a) indicate vector norms. In contrast, the outer norms indicate the operator norm of the matrix indexed by star $(*)$. More specifically, $||A(*,*)||$ refers to the operator norm of the matrix with matrix elements $A_{ij}$. Thus $|| ||\kappa(\boldsymbol{x}*,\cdot*)|| ||$ is the operator norm $||A||$ of the matrix $A$ with matrix elements $A_{ij} = ||\kappa(\boldsymbol{x}i, j\cdot)||$. $\kappa(\boldsymbol{x}b_1, a_2b_2)$ denotes $\sum_{a_1} \kappa(a_1b_1, a_2b_2)x_{a_1}$, where $\boldsymbol{x}$ is a vector.*

We do not want to explain the cumulant norms beyond what has been said, considering it is too technically involved and rather a distraction. For interested readers, we suggest reading the paper Erdős et al. (2017). Equipped with the cumulant norms, we would have the assumptions stated in assumption 5.1 5.2 that make MDE valid.

**Remark.** *In Erdős et al. (2017), the functions $f, g_1, \dots, g_q$ in assumption 5.2 are assumed to be functions without any qualifiers. We change it to analytic functions for further usage. In the proof of theorem F.1, the functions are only required to be analytic, thus even if the assumptions are changed, the conclusion still holds.*

The diversity assumption requires that a matrix entry $w_\alpha$ only couples with a minority of the overall entries, and for the rest of the entries, the coupling strength does not exceed a certain value $N^{-3q}||f||_{q+1} \prod_{j=1}^q ||g_j||_{q+1}$ characterized by cumulants. For example, suppose $q = 1$, given a entry $\alpha_1$, the assumption essentially states that the entries in the coupling set $\mathcal{N}(\alpha_1)$ is not strongly coupled with the resting of the population $\boldsymbol{W}_{\mathbb{I} \setminus \mathcal{N}_{n_1+1}(\alpha_1)}$. The explanation goes similar as $q$ grows, of which the coupling strength is characterized by higher order cumulants. While boundedness assumptions 1)2)3)4) states the expectation of $\boldsymbol{H}$ is bounded, moments are finite, cumulants are bounded for the entries that do strongly couples, and $\mathcal{S}[\boldsymbol{W}]$ is bounded in the sense of eigenvalues.

When the assumptions satisfies, we have the resolvent $\boldsymbol{G}$ of a random matrix $\boldsymbol{H}$ close to the solution to the MDE probabilistically with some regular properties as the following, adopted in an informal style to ease reading from Erdős et al. (2017) theorem 2.2, Helton et al. (2007) theorem 2.1, and Alt et al. (2018) theorem 2.5.

**Theorem F.1.** *Let $\boldsymbol{M}$ be the solution to the Matrix Dyson Equation eq. (9), and $\rho$ the density function (measure) recovered from normalized trace $\frac{1}{N}tr\, \boldsymbol{M}$ through Stieljies inverse lemma F.1. We have*

*1. The MDE has a unique solution $\boldsymbol{M} = \boldsymbol{M}(z)$ for all $z \in \mathbb{H}$.*

2. $supp\rho$ *is a finite union of closed intervals with nonempty interior. Moreover, the nonempty interiors are called the* **bulk** *of the eigenvalue density function* $\rho$.

3. *The resolvent* $\boldsymbol{G}$ *of* $\boldsymbol{H}$ *converges to* $\boldsymbol{M}$ *as* $N \to \infty$.

**Remark.** *theorem F.1.3 is a probably-approximately-correct type result, where the error depends on* $N$. *We do not present the exact error bound here, for that it is rather complicated, and does not help understanding — since we are not working on finer behaviors of NNs with a particular size, and do not need such a fine granularity characterization yet. We refer interested readers to Erdős et al. (2017) theorem 2.1, 2.2, where the exact error bounds are present.*

### F.3 DIVERSITY ASSUMPTION IS A PRECONDITION TO THE POWER OF NNS AND S-SYSTEM

Before we leverage MDE to obtain the eigen-spectrum of the Hessian $\boldsymbol{H}$ of NNs derived at eq. (5), we explain the meaning of the assumptions 5.1, 5.2 in the NN and S-System context, so to point to the potential of the assumptions to give practical guidance on training NNs.

Recall that the objective function is the empirical risk function $R(T)$ at eq. (1). Given a set of i.i.d. training samples $(X_i, Y_i)_{i=1,\dots,m}$, $R(T)$ is a summation of the i.i.d. random matrices. Formally, reusing the notation to denote $\boldsymbol{H}$ the Hessian of $R(T)$ and $\boldsymbol{H}_i$ the Hessian of $l(T(X_i; \boldsymbol{\theta}), Y_i)$, we have

$$\boldsymbol{H} = \frac{1}{m} \sum_{i=1}^{m} \boldsymbol{H}_i \tag{12}$$

Acute reader may realize that by the multivariate central limiting theorem (Klenke (2012) theorem 15.57), $\boldsymbol{H}$ will converge to a Gaussian ensemble, i.e., a random matrix of which the distribution of entries is a Gaussian process (GP), asymptotically as $m \to \infty$. For a GP, all higher order cumulants are zero, which greatly simplifies the picture, and gives much clearer meaning on the assumptions 5.1 5.2 made. In the following, we will explain the practicality, and also how they may serve as guidance to design and improve NNs, in the asymptotically large sample limit, which gives a picture that can be described using more widely used terms, i.e., mean and covariance.

The practicality of boundedness assumptions is obvious, since we do not want values to blow up. We only note for the two outliers. First, the lower bound in 4) in boundedness assumption, $cN^{-1}\text{tr}\,\boldsymbol{G} \preceq \mathcal{S}[\boldsymbol{G}]$, which is not about infinity. It asks the eigenvalues of $\mathcal{S}[\boldsymbol{G}]$ to stay close to its average value, so to let $\mathcal{S}[\boldsymbol{G}]$ stay in the cone of the positive definite matrices to ensure the stability of the MDE. It is essentially a constraint on the interaction of second order cumulants, and is realizable in a NN, though we are not clear on its physical meaning for the time being. Second, the boundedness assumption 3), as briefly discussed before, is a bound that bounds the strength of the entries of $\boldsymbol{H}$ that do correlate, while the diversity assumption is about the weakness of the entries that are not correlated. The practicality of the former is straightforward. To see the practicality of diversity assumption in the NN context, first we come back to the concrete form of Hessian. We rewrite eq. (4) in the following form (the equation should be read vertically)

$$\boldsymbol{H}_{pq} = l'(T\boldsymbol{x}, y)\tilde{\boldsymbol{W}}_{q\sim(L-1)}\text{dg}(\tilde{\boldsymbol{h}}''_{L-1})\boldsymbol{\alpha} \qquad\qquad = \tilde{\boldsymbol{\alpha}}_q \tag{13a}$$

$$\otimes \tilde{\boldsymbol{W}}_{p\sim(q-1)} \qquad\qquad \otimes \tilde{\boldsymbol{W}}_{p\sim(q-1)} \tag{13b}$$

$$\otimes \boldsymbol{x}^T\tilde{\boldsymbol{W}}_{1\sim(p-1)} \qquad\qquad \otimes \tilde{\boldsymbol{x}}^T_{p-1} \tag{13c}$$

where

$$\tilde{\boldsymbol{W}}_{q\sim(L-1)} := \text{dg}(\tilde{\boldsymbol{h}}''_q)\prod_{k=q+1}^{L-2}(\boldsymbol{W}_k\text{dg}(\tilde{\boldsymbol{h}}''_k))\boldsymbol{W}_{L-1}, \quad \tilde{\boldsymbol{\alpha}}_q := \tilde{\boldsymbol{W}}_{q\sim(L-1)}\text{dg}(\tilde{\boldsymbol{h}}''_{L-1})\boldsymbol{\alpha}l'(T\boldsymbol{x}, y)$$

$$\tilde{\boldsymbol{W}}_{p\sim(q-1)} := \prod_{j=p+1}^{q-1}(\text{dg}(\tilde{\boldsymbol{h}}'_j)\boldsymbol{W}_j^T)\text{dg}(\tilde{\boldsymbol{h}}'_p),$$

$$\tilde{\boldsymbol{W}}_{1\sim(p-1)} := \prod_{i=1}^{p-1}(\boldsymbol{W}_i\text{dg}(\tilde{\boldsymbol{h}}_i)), \qquad\qquad \tilde{\boldsymbol{x}}^T_{p-1} := \boldsymbol{x}^T\tilde{\boldsymbol{W}}_{1\sim(p-1)}$$

With some efforts, using NN terminologies, it can be viewed that eq. (13a) is a vector $\tilde{\boldsymbol{\alpha}}_q$ created by back propagating the vector $\text{dg}(\tilde{\boldsymbol{h}}''_{L-1})\boldsymbol{\alpha}l'(T\boldsymbol{x}, y)$ to layer $q$ by left multiplying $\tilde{\boldsymbol{W}}_{q\sim(L-1)}$— note

that if you replace $\boldsymbol{h}_k''$ with $\boldsymbol{h}_k'$, you get the *back propagated gradient*; eq. (13b) is the *covariance matrix without removing the mean* between neurons at layer $p$ and layer $q-1$, when taking expectation w.r.t. samples, i.e., $\mathbb{E}_{z\sim\mu^z}[\tilde{\boldsymbol{W}}_{p\sim(q-1)}]$; eq. (13c) is the *forward propagated activation* at layer $p-1$. Now it is quite clear what the correlation between entries of the Hessian is about. It is the correlation between *the product of forward propagated neuron activation at layer $p-1$, the back propagated "gradient" at layer q, and the strength of activation paths that connects the two sets of neurons.*

When $\boldsymbol{H}$ is a Gaussian ensemble, all higher order cumulants vanishes, thus the diversity assumption is solely about the second order cumulants, and the case when $q=1$ and $q=2$. Since $f,\{g_i\}_{i=1,...,q}$ are analytic, $\kappa(f(\cdot),g_1(\cdot))$ is a generalized cumulant (McCullagh (1987) Chapter 3), which can be decomposed into a sum of cumulants of entries of the Hessian. So is $\kappa(f(\cdot),g_1(\cdot),g_2(\cdot))$. Considering that only first and second cumulants exist, which are means and covariance respectively, $\kappa(f(\cdot),g_1(\cdot))$ thus is a sum of means of entries of the Hessian and covariance between entries of the Hessian. So is $\kappa(f(\cdot),g_1(\cdot),g_2(\cdot))$. Using $\kappa(f(\cdot),g_1(\cdot))$ as an example, the diversity assumption states that for any $\alpha\in\mathbb{I}$, nested sets $\mathcal{N}_1\subset\mathcal{N}_2=\mathcal{N},|\mathcal{N}|\leq N^{1/2-\mu}$ exist (when only cumulants up to the second order exist, it suffices to let $R$ be 2 instead of every $R\in\mathbb{N}$ (Erdős et al. (2017))), such that $\kappa(f(\boldsymbol{W}_{\mathbb{I}\setminus\mathcal{N}}),g_1(\boldsymbol{W}_{\mathcal{N}_1}))=\sum_{\beta\in\mathcal{N}_1'\subset\mathcal{N}_1\cup\mathbb{I}\setminus\mathcal{N}}\kappa(\beta)+\sum_{\beta,\gamma\in\mathcal{N}'\subset\mathcal{N}_1\cup\mathbb{I}\setminus\mathcal{N}}\kappa(\beta,\gamma)$, where $\mathcal{N}_1',\mathcal{N}$ are subsets noted that depends on $f,g_1$. The interpretation is qualitative. But even from the qualitative interpretation, it can see that *the diversity assumption is on the smallness of the mean and covariance of $\tilde{\alpha}_i\tilde{w}_{jk}\tilde{x}_l$, the product of "gradient", activation path correlation strength, and forward activation.* Additionally, to prove theorem 5.1, we need a further assumption that $\mathbb{E}[\boldsymbol{H}]=\boldsymbol{0}$. It clearly connects to the experiment tricks used in the community, such as the early initialization schemes that tries to keep mean of gradient and activation zero, and standard deviation (std) small (Glorot & Bengio (2010) He et al. (2015)), the normalization schemes that keep the mean of activation zero and std small (Ioffe, Sergey and Szegedy (2015)Salimans & Kingma (2016)), though some more works are needed to reach there rigorously.

To recap, as stated similarly in section 2, appendix F.2, the diversity assumption states that a diversity should exist in the neuron population, so that for any neurons, it does not strongly correlate with the majority of the neuron population. The diversity in r.e.s. of S-System is not a built-in feature, but a design choice in its implementations.

Qualitatively, we can see the design of NNs resonates with the diversity assumption: 1) different group indicator r.e.s. are assumed to be independent given input r.e.s., referring to definition 4 5, in which case, the coupling aims to group the measure that is distinctive w.r.t. other couplings created through grouping, thus, r.e.s that are not coupled together are likely to be uncorrelated; 2) activation function creates couplings that only couple higher scale events with the "active" lower scale events, thus implementing coarse graining that creates events that are composed by different lower scale events. The above design may not be the only choice, however, it helps create uncorrelated r.e.s within a scale and across scales, consequently making the product of the forward propagated activation, activation paths, and back propagated "gradient" tend to be uncorrelated.

Yet, this is a rather general explanation on why diversity occurs without taking into the finer statistics structure in the data. More improvements may still be made. For instance, the low correlation existed in CNN is the result of a coupling that considers the spatial symmetry, where output r.e.s in a large spatial distance simply does not couples, thus tending to be uncorrelated.

S-System is a fabulous mechanism that can indefinitely increase the number of parameters, thus its learning capacity, in a meaningful way, i.e., creating higher scale coupling yet maintaining the diversity of the r.e.s created. Such mechanism does not normally hold in other systems or algorithms. Taking linear NNs for example, though with the potential to infinitely increase its parameters, matrices that multiply together still have a highly correlation structure within, thus cannot create a population of diverse neurons that are of low correlation with a majority of the other neurons. Accompanying the result we will prove in the next section, which states $R(T)$ can be optimized to zero, assuming assumptions 5.1 5.2, we can see that *the diversity assumption actually characterizes a sufficient precondition to the optimization power of NNs.*

### F.4 NN LOSS LANDSCAPE: ALL LOCAL MINIMA ARE GLOBAL MINIMA WITH ZERO LOSSES

We have obtained the operator equation to describe the eigen-spectrum of the Hessian of NNs and explained its assumptions. With one further assumption, we show in this section that for NNs with objective function belonging to the function class $\mathcal{L}_0$, all local minima are global minima with zero loss values.

We outline the strategy first. Since MDE is a nonlinear operator equation, it is not possible to obtain a close form analytic solution. The only way to get its solution is an iterative algorithm (Helton et al. (2007)), which is not an easy task given the millions of parameters of a NN — remembering that we are dealing with large $N$ limit — though it can serve as an exploratory tool. However, we do are able to get qualitative results by directly analyzing the equation. Our goal is to show all critical points are saddle points, except for the ones has zero loss values, which are global minima. To prove it, we prove that at the points where $R(T) \neq 0$, the eigen-spectrum $\mu_H$ of the Hessian $H$ is symmetric w.r.t. the $y$-axis, which implies that as long as non-zero eigenvalues exist, half of them will be negative. To prove it, we prove the stieltjes transform $m_{\mu_H}(z)$ of $\mu_H$ satisfies $\Im m_{\mu_H}(-z^*) = \Im m_{\mu_H}(z)$, where $z^*$ denote the complex conjugate of $z$. In the following, we present the proof formally.

**Lemma F.2.** *Let $M(z), M'(-z^*)$ be the unique solution to the MDE at spectral parameter $z, -z^*$ defined at eq. (9) respectively, and $A = 0$. We have*

$$M' = -M^*$$

*where $*$ means taking conjugate transpose.*

*Proof.* First, we rewrite the MDE. Note that $\mathcal{S}[G]$ is positivity preserving, i.e., $\forall G \succ 0, \mathcal{S}[G] \succ 0$ by assumption 5.1 4). In addition, we have $\Im z > 0$, thus $\Im(z + \mathcal{S}[G]) \succ 0$. Then, by Haagerup & Thorbjørnsen (2005) lemma 3.2, we have $z + \mathcal{S}[G] \succ 0$, so it is invertable. Thus, we can rewrite the MDE into the following form

$$G = -(z + \mathcal{S}[G])^{-1} \tag{14}$$

Suppose $M$ is a solution to the MDE at spectral parameter $z$. The key to the proof is the fact that $\mathcal{S}[G]$ is linear and commutes with taking conjugate, thus by replacing $M$ with $-M^*$, and $z$ with $-z^*$, we would get the same equation. We show it formally in the following.

First, note that $\mathcal{S}[M]$ is a linear map of $M$, so the we have

$$\mathcal{S}[-M] = -\mathcal{S}[M]$$

Also, $\mathcal{S}[M]$ commutes with $*$, for the fact

$$\mathcal{S}[M^*] = \mathbb{E}[W M^* W] = \mathbb{E}[(W M W)^*] = \mathbb{E}[W M W]^* = \mathcal{S}[M]^*$$

Furthermore, we $*$ is commute with taking inverse, for the fact

$$AA^{-1} = I$$
$$\implies (AA^{-1})^* = I$$
$$\implies A^{-1*}A^* = I$$
$$\implies A^{-1*} = A^{*-1}$$

With the commutativity results, we do the proof. The solution $M$ satisfies the equation

$$M = -(z + \mathcal{S}[M])^{-1}$$

Replacing $M$ with $-M^*$, $z$ with $-z^*$, we have

$$-M^* = -(-z^* + \mathcal{S}[-M^*])^{-1}$$
$$\implies M^* = -(z^* + \mathcal{S}[M^*])^{-1}$$
$$\implies M^* = -(z^* + \mathcal{S}[M]^*)^{-1}$$
$$\implies M^* = -(z + \mathcal{S}[M])^{-1*}$$
$$\implies M = -(z + \mathcal{S}[M])^{-1}$$

After the replacement, we actually get the same equation. Thus, $-\boldsymbol{M}^*, -z^*$ also satisfy eq. (14). Since the pair also satisfies the constrains $\Im \boldsymbol{M} \succ 0, \Im z > 0$, and by theorem F.1, the solution is unique, we proved the solution $\boldsymbol{M}'$ at the spectral parameter $-z^*$ is $-\boldsymbol{M}^*$.

□

**Theorem F.2.** *Let $\boldsymbol{H}$ be a real symmetric random matrix satisfies assumptions 5.1 5.2, in addition to the assumption that $\boldsymbol{A} = \boldsymbol{0}$. Let the ESD of $\boldsymbol{H}$ be $\mu_{\boldsymbol{H}}$. Then, $\mu_{\boldsymbol{H}}$ is symmetric w.r.t. to y-axis. In other words, half of the non-zero eigenvalues are negative. Furthermore, non-zero eigenvalues always exist, implying $\boldsymbol{H}$ will always have negative eigenvalues.*

*Proof.* By theorem F.1, the resolvent $\boldsymbol{G}$ of $\boldsymbol{H}$ is given by the the unique solution to eq. (9) at spectral parameter $z$. Let the solution to eq. (9) at spectral parameter $z, -z^*$ be $\boldsymbol{M}, \boldsymbol{M}'$, By lemma F.2, we have the solutions satisfies

$$\boldsymbol{M}' = -\boldsymbol{M}^*$$

By F.1, the ESD of $\boldsymbol{H}$ at $\Re z$ is given at

$$\mu_{\boldsymbol{H}}(\Re z) = \lim_{\Im z \to 0} \frac{1}{\pi} \Im m_{\mu_{\boldsymbol{H}}}(z)$$

Since $m_{\mu_{\boldsymbol{H}}}(z) = \frac{1}{N} \mathrm{tr}\, M$, we have

$$\mu_{\boldsymbol{H}}(\Re z) = \lim_{\Im z \to 0} \frac{1}{\pi} \frac{1}{N} \Im \mathrm{tr}\, M$$

Similarly,

$$\mu_{\boldsymbol{H}}(\Re(-z^*)) = \lim_{\Im(-z^*) \to 0} \frac{1}{\pi} \frac{1}{N} \Im \mathrm{tr}\, M'$$

Note that

$$\mu_{\boldsymbol{H}}(\Re(-z^*)) = \lim_{\Im(-z^*) \to 0} \frac{1}{\pi} \frac{1}{N} \Im \mathrm{tr}\, M'$$

$$\implies \mu_{\boldsymbol{H}}(\Re(-z^*)) = \lim_{\Im(-z^*) \to 0} \frac{1}{\pi} \frac{1}{N} \Im \mathrm{tr}\, (-M^*)$$

$$\implies \mu_{\boldsymbol{H}}(-\Re z) = \lim_{\Im z \to 0} \frac{1}{\pi} \frac{1}{N} \Im \mathrm{tr}\, M$$

Thus, $\mu_{\boldsymbol{H}}(\lambda), \lambda \in \mathbb{R}$ is symmetric w.r.t. $y$-axis. It follows that for all non-zero eigenvalues, half of them are negative.

By theorem F.1 2, there are always bulks in $\mathrm{supp}\mu_{\boldsymbol{H}}$, thus there are always non-zero eigenvalues. Since half of the non-zero eigenvalues are negative, it follows $\boldsymbol{H}$ always has negative eigenvalues.

□

*Proof of theorem 5.1.* First, we prove part 1 of the theorem. The majority of the proof of part 1 have been dispersed earlier in the paper. What the proof here does mostly is to collect them into one piece.

The Hessian $\boldsymbol{H}$ of the risk function eq. (1), can be decomposed into a summation of Hessians of loss functions of each training sample, which is described in eq. (12). For each Hessian in the decomposition, it is computed in eq. (5), and it has been shown that $\boldsymbol{H}$ is a random matrix in section 5 and appendix F.1.

The analysis of the random matrix $\boldsymbol{H}$ needs to break down into two cases: 1) for all training samples, at least one sample $(x, y)$ has non-zero loss value; 2) and all training samples are classified properly with zero loss values.

We first analyze case 1), since the loss $l$ belongs to function class $\mathcal{L}_0$, $l$ is convex and is valued zero at its minimum. When $l(x, y) \neq 0$, we have $l'(x, y) \neq 0$, thus $\boldsymbol{H}$ is a random matrix — not a zero matrix. The analysis of this type of random matrix is undertaken in appendix F.2. For a NN, the assumptions 5.1 5.2 can be satisfied, and the eigen-spectrum $\mu_{\boldsymbol{H}}$ of $\boldsymbol{H}$ is given by the MDE defined at eq. (9). The practicality and its potential to guide real world NN optimization is discussed in appendix F.3.

By theorem F.2, $\mu_{\boldsymbol{H}}$ is symmetric w.r.t. $y$-axis, and half of its non-zero eigenvalues are negative. Thus, for all critical points of $R(T)$, its will have half of its non-zero eigenvalues negative. It implies all critical points are saddle points.

Now we turn to the case 2). In this case, all training samples are properly classified with zero loss value. Considering the lower bound of $l$ is zero, we have reached the global minima. Also, since all critical points in case 1) are saddle points, local minima can only be reached in case 2), implying all local minima are global minima. Thus, the first part of the theorem is proved.

Now we prove part 2 of the theorem.

Note that the minima is reached for the fact that we have reached the situation where the Hessian $\boldsymbol{H}$ has degenerated into a zero matrix. Thus, each local minimum is not a critical point, but an infimum, where in a local region around the infimum in the parameter space, all the eigenvalues are increasingly close to zero as the parameters of the NN approach the parameters at the infimum. We show it formally in the following.

Writing a block $\boldsymbol{H}_{pq}$ (defined at eq. (4)) in the Hessian $\boldsymbol{H}_i$ of one sample (defined at eq. (5)) in the form of

$$\boldsymbol{H}_{pq} = l'(T\boldsymbol{x}, y)\tilde{\boldsymbol{H}}_{pq}$$

where $i$ is the index of the training samples, defined at eq. (1). Then, putting together $\tilde{\boldsymbol{H}}_{pq}$ together to form $\tilde{\boldsymbol{H}}_i$, $\boldsymbol{H}_i$ is rewritten in the form of

$$\boldsymbol{H}_i = l'(T\boldsymbol{x}_i, y_i)\tilde{\boldsymbol{H}}_i$$

Then the Hessian $\boldsymbol{H}$ (defined at eq. (12)) of the risk function defined eq. (1) can be rewritten in the form of

$$\boldsymbol{H} = \frac{1}{m}\sum_{i=1}^{m} l'(T\boldsymbol{x}_i, y_i)\tilde{\boldsymbol{H}}_i$$

Taking the operator norm on the both sides

$$||\boldsymbol{H}|| = ||\frac{1}{m}\sum_{i=1}^{m} l'(T\boldsymbol{x}_i, y_i)\tilde{\boldsymbol{H}}_i|| \leq \frac{1}{m}\sum_{i=1}^{m}|l'(T\boldsymbol{x}_i, y_i)|\,||\tilde{\boldsymbol{H}}_i||$$

Denote $\max_i\{||\tilde{\boldsymbol{H}}_i||\}$ as $\lambda_0$, we have

$$||\boldsymbol{H}|| \leq \frac{1}{m}\sum_{i=1}^{m}|l'(T\boldsymbol{x}_i, y_i)|\lambda_0$$
$$= \mathbb{E}_m[l'(TX, Y)]\lambda_0$$

The above inequality shows that, as the risk decreases, more and more samples will have zero loss value, consequently $l' = 0$, thus $\mathbb{E}_m[l']$ will be increasingly small, thus the operator norm of $\boldsymbol{H}$. At the minima where all $l' = 0$, the Hessian degenerates to a zero matrix. $\qquad\square$

**Remark.** *It is not necessary for the assumption $\boldsymbol{A} = \boldsymbol{0}$ to be held for the theorem to hold, or more specifically, for $\boldsymbol{H}$ to have negative eigenvalues at its critical points. By proposition 2.1 in Alt et al. (2018)*

$$supp\mu_{\boldsymbol{H}} \subset Spec\boldsymbol{A} + [-2||\mathcal{S}||^{1/2}, 2||\mathcal{S}||^{1/2}]$$

*where $Spec\boldsymbol{A}$ denotes the support of the spectrum of the expectation matrix $\boldsymbol{A}$ and $||\mathcal{S}||$ denotes the norm induced by the operator norm. Thus, it is possible for the spectrum $\mu_{\boldsymbol{H}}$ of $\boldsymbol{H}$ to lie at the left side of the $y$-axis, as long as the spectrum of $\boldsymbol{A}$ is not too way off from the origin. However, existing characterizations on $supp\mu_{\boldsymbol{H}}$ based on bound are too inexact to make sure the existence of support on the left of the $y$-axis. To get rid of the zero expectation assumption, more works are needed to obtain a better characterization, and could be a direction for future work.*

The the phenomenon characterized by theorem 5.1.1 is rather remarkable, if not marvelous. It shows that instead of seeing non-convex optimization as something to avoid, a class of non-convex objective functions can be that powerful to the point of "solving" — minimizing the error to the point of vanishing — complex problems that nature is dealing with in a rather reliable fashion. We feel like this is how a brain is doing optimization. We envision that a much larger class of functions

possess such benign loss landscapes than the one here we have studied. Actually, we have isolated a function class that represents some of the most essential characteristics of a more general class of function as shown in eq. (2), so that we can show the principle underlying. That is, diverse yet cooperative subunits aggregating together to form a system can optimize an objective consistently. This larger class of function could be as important as the concept of convexity, and would play an important role in optimization. The goal of the paper is to lay the backbone of the theory of the NNs that make the principles underlying clear, instead of presenting the theory in its complete form in one go. Thus, essential properties of the function class are yet to be identified, and will be part of our future work.

The theorem also contributes to explaining why depth is crucial. The large $N$ limits of the Hessian can be achieved by adding more layers (in the terminology of S-System, using a scale poset having a longer chain as a subset), even though the number of neurons in each of the layers may be quite small compared with the overall number of neurons. The diversity of neurons is possible due to activation functions (in the terminology of S-System, conditional grouping extension on estimated realizations of previous created output r.e.s.).

The phenomenon characterized by theorem 5.1.2 explains why there are two phases in the training dynamics of NNs, i.e., the rapid error decreasing phase when loss value is high, and the slow error decaying phase when the loss value is close to minima. As the error decreases, the expectation of the derivative of loss values in $R(T)$ will increasingly approach zero, thus the $\text{supp}\mu_{\boldsymbol{H}}$ will concentrate around zero increasingly, making the landscape increasingly flat and the training process slowly. It probably also explains why we need to gradually decrease the step size in the gradient descent algorithms in practice. Very likely the flat regions are of a small volume compared with the overall parameter space. Thus, if the training goes conservatively, and inches towards the global minima, the risk will gradually decrease. But if we give a powerful kick to the training that induces a large shift in the parameter space, it may kick the current parameter out of the flat region that can inch toward the global minima, like kicking a ball from a valley to another mountain in the hyperspace, thus making the training starts all over again to find a valley to decrease the risk. A further characterization of the landscape goes beyond infinitesimal local regions may rigorously prove the conjecture. It even poses the possibility to move across the flat region rather swiftly, as long as we figure out how to stay in the valley as we stride big.

# G    LEARNING FRAMEWORK OF S-SYSTEM

As described in the problem formulation in section 5, supervised statistical learning is to minimize the discrepancy between the approximated measure and the empirical measure. Created by S-System, the approximated measure is to approximate the measure of a group of events in the PPMS. Thus, supervised learning in S-System trained by gradient descent through back propagation (BP) already has a solid theoretical foundation with well explained behaviors. Yet, it is not the complete picture. We present here initially the learning framework of S-System, of which supervised learning and unsupervised learning are two special cases.

Let $\mathcal{M}^{\mathcal{W}}$ be an event representation built by an S-system with scale poset $\mathcal{S}^{\mathcal{Z}}$ built on a measurement collection r.e. $Z := (X, Y)$ with measure $\mu^{\mathcal{Z}}$ supported on the PPMS $\mathcal{W}$. The measurable space $\mathcal{Z}$ is a product space $\mathcal{X} \times \mathcal{Y}$, where $\mathcal{X}$ denotes the data space, and $\mathcal{Y}$ denotes label space. Let $\mathcal{M}_s$ be the event representation built at scale $s$. Let $\mu_s^{\mathcal{H}}, s \in \mathcal{S}^{\mathcal{Z}}$ be the probability measure on output r.e.s $(H_s, \hat{H}_s)$ of $\mathcal{M}_s$, and $\nu_s$ the law on conditional group indicator r.e.. The learning of S-System is to minimize the discrepancy between measure $\mu^{\mathcal{W}}(X^{-1}(W_s^{-1}(A)))$ and $\mu_s^{\mathcal{H}}(A)$ assigned to a event $A \subset \Omega^{\mathcal{H}_s^e}$, where $\Omega^{\mathcal{H}_s^e}$ is the event space of the probability measure space of $\mathcal{M}_s$, and $W_s$ is the transport map of the coupling probability kernel of $\mathcal{M}_s$. One way to characterize the discrepancy is Maximum Likelihood Estimation (MLE), where the parameters that most likely to generate the data

consist the best estimator. The likelihood function of $\mathcal{M}^{\bar{\mathcal{W}}}$ is

$$p(X; \boldsymbol{\theta}) = \sum_{\mathcal{O}^{\bar{\mathcal{W}}} \setminus X} p(\mathcal{O}^{\bar{\mathcal{W}}}; \boldsymbol{\theta})$$

$$= \prod_{s_L \in \mathcal{S}_L} \sum_{\boldsymbol{h}_{s_L}} \int_{\hat{\boldsymbol{h}}_{s_L}} \sum_{\boldsymbol{h}_{s'}, s' \in p(s_L)} \int_{\hat{\boldsymbol{h}}_{s'}, s' \in p(s_L)} \sum_{\boldsymbol{h}_{s''}, s'' \in p(s')} \int_{\hat{\boldsymbol{h}}_{s''}, s'' \in p(s')} \cdots$$

$$\mu_{s_L}^{\mathcal{H}}(H_{s_L}, \hat{H}_{s_L} | H_{p(s_L)}, \hat{H}_{p(s_L)}) \prod_{s' \in p(s_L)} \mu_{s'}^{\mathcal{H}}(H_{s'}, \hat{H}_{s'} | H_{p(s')}, \hat{H}_{p(s')}) \prod_{s'' \in p(s')} \cdots \mu^{\mathcal{Z}}(X)$$

where $\mathcal{O}^{\bar{\mathcal{W}}}$ and $\boldsymbol{\theta}$ are the r.e. set and the parameters of $\mathcal{M}^{\bar{\mathcal{W}}}$ respectively, $\mathcal{S}_L$ denotes the set of largest element in $\mathcal{S}^{\mathcal{Z}}$, and $p(s)$ denotes the elements in $\mathcal{S}^{\mathcal{Z}}$ that are the predecessors of $s$. Depending on whether $H_s, \hat{H}_s$ are discrete or not, the summation may be changed to integral, vice versus. It could be understood as getting the marginal probability distribution of $X$ from a factorized probability of a direct acyclic graph in probabilistic graphical model (PGM) (Koller & Friedman (2009)).

Needless to say, the likelihood function is intractable when the r.e. set $\mathcal{O}^{\bar{\mathcal{W}}}$ gets large. Perhaps more importantly, we do not know $\mu^{\mathcal{Z}}(X)$, so we do not know $\mu_s^{\mathcal{H}}$ since it is built on the transport map applied on $X$. Thus, to make the estimation tractable, and to faithfully estimate measure on events groups already seen without making assumptions on $\mu^{\mathcal{Z}}(X)$, we make the following decomposition of the log likelihood function to focus on estimating measures on group indicators r.e.s

$$\ln p(\mathcal{O}^{\bar{\mathcal{W}}}; \boldsymbol{\theta}) = \mathcal{L}(\{\nu_s\}_{s \in \mathcal{S}^{\mathcal{Z}}}, \boldsymbol{\theta}) + \sum_{s \in \mathcal{S}^{\mathcal{Z}}} D_{\text{KL}}^s(\nu_s || \mu^{\mathcal{X}}(H_s | W_s(X)))$$

where

$$\mathcal{L}(\{\nu_s\}_{s \in \mathcal{S}^{\mathcal{Z}}}, \boldsymbol{\theta}) = \sum_{\boldsymbol{h}_s, s \in \mathcal{O}^{\bar{\mathcal{W}}} \setminus X} q(\mathcal{O}^{\bar{\mathcal{W}}} \setminus X) \ln \frac{\mu^{\mathcal{Z}}(X)}{q(\mathcal{O}^{\bar{\mathcal{W}}} \setminus X)}$$

$$q(\mathcal{O}^{\bar{\mathcal{W}}} \setminus X) = \prod_{s_L \in \mathcal{S}_L} \mu_{s_L}^{\mathcal{H}}(H_{s_L} | \hat{H}_{s_L}) \prod_{s' \in p(s_L)} \mu_{s'}^{\mathcal{H}}(H_{s'} | \hat{H}_{s'}) \prod_{s'' \in p(s')} \cdots$$

$$= \prod_{s_L \in \mathcal{S}_L} \nu_{s_L} \prod_{s' \in p(s_L)} \nu_{s'} \prod_{s'' \in p(s')} \cdots$$

$$D_{\text{KL}}^s(\nu_s || \mu^{\mathcal{X}}(H_s | W_s(X))) = \sum_{H_s} \nu_s \ln \frac{\mu^{\mathcal{X}}(H_s | W_s(X))}{\nu_s}$$

Note that $\mu_s^{\mathcal{H}}(H_s | \hat{H}_s)$ is used instead of $\mu_s^{\mathcal{H}}(H_s | \hat{H}_s, \hat{H}_{p(s)})$ because in CGE, $H_s$ is conditional independent with previous output r.e.s given $\hat{H}_s$. $\mathcal{L}(\{\nu_s\}_{s \in \mathcal{S}^{\mathcal{Z}}}, \boldsymbol{\theta})$ is called *expected data log likelihood*, $\ln p(\mathcal{O}^{\bar{\mathcal{W}}}; \boldsymbol{\theta})$ the *complete data log likelihood* and $D_{\text{KL}}^s(\nu_s || \mu^{\mathcal{X}}(H_s | W_s(X)))$ is the *KL divergence at scale $s$* between estimated measure and true measure.

The decomposition has been used widely in PGM (Bishop (2006)). Successful techniques derived from it have been invented known as Variation Inference and Expectation Propagation etc. Yet, one remarkable difference in the above decomposition and existing decomposition is that here we decompose the probability measure on physical events in APMS $\bar{\mathcal{W}}$, and estimate measure $\nu_s$ that aims to approximate the measure of groups of events in the event space of the PPMS $\mathcal{W}$, while in existing decomposition, their approaches are to hallucinate some parametric probabilistic models on $\mathcal{O}^{\bar{\mathcal{W}}}$ (under the context of S-System), and because the "exact" inference is intractable, they use the decomposition to make the inference tractable. In essence, we are not making any assumptions on $Z$, but only on how they are supposed to group together, while existing approaches using the decomposition is solely about making assumptions on $Z$, and how to make the computation tractable, thus likely leading to significant model biases.

With the above decomposition, we can see what the training of a supervised NN is. Forward propagation (FP) is to estimate values of group indicator r.e.s by assigning $H_s$ a value that maximizes the expected data log likelihood $\mathcal{L}(\{\nu_s\}_{s \in \mathcal{S}^{\mathcal{Z}}}, \boldsymbol{\theta})$ w.r.t. $q(\mathcal{O}^{\bar{\mathcal{W}}} \setminus X)$ through activation function (though depending on the activation function chosen, it may not always reach the maximum), while holding

$\boldsymbol{\theta}$ fixed. BP is to minimize the KL divergence $D_{\mathrm{KL}}^s$ at scale $s$ w.r.t. $\boldsymbol{\theta}$ whenever there is a supervisory information/label on $H_s$ supervising how the events are supposed to group, while holding $q(\mathcal{O}^{\mathcal{W}} \setminus X)$ fixed.

The decomposition not only includes supervised NNs, but also includes variational autoencoder (VAE) (Welling (2014)), where further assumptions on probability measure of $X$ are assumed. When absent of labels, a normal distribution on $H_s$ is assumed, thus encouraging each group indicator r.e. to learn a grouping that is supposed to be disentangled with the rest. When some labels exist, we recover semi-supervised VAE.

Thus, supervised learning is never something that stands on its own, so is unsupervised learning. They are two perspectives to look at the same thing, or they are Yin and Yang of the Tao in Chinese philosophy, or the thesis and anti-thesis of dialectics. They are different ways with different assumptions to get information to approximate the measure of events groups, e.g., the group indicator r.e.s in S-System, which represents what has been recognized. Even pure supervised learning can do some unsupervised learning — by pushing KL divergence $D_{\mathrm{KL}}^s$ at some scales to zero, the expected data log likelihood will be closer to the complete data log likelihood. This partly explains the emergence of generic feature in NNs (though the maximization of $\mathcal{L}(\{\nu_s\}_{s \in \mathcal{S}^{\mathcal{Z}}}, \boldsymbol{\theta})$ perhaps is the main reason). So pure unsupervised learning can do some supervised learning — the maximization of $\mathcal{L}(\{\nu_s\}_{s \in \mathcal{S}^{\mathcal{Z}}}, \boldsymbol{\theta})$ leads to a smaller KL divergence. We do not observe it in an obvious way in experiments because the grouping does not necessarily concur to the grouping we humans already have. By imposing some structure on the grouping scheme, e.g., imposing a normal distribution, we can discover manifolds that groups events that make sense, e.g., facial expression or digit variations (Welling (2014)).

Lastly, we note Bayesian aspects can be further included in the learning framework by endowing assumptions further on the parameter space.

## H  RELATED WORKS

### H.1  HIERARCHY

The idea that the data space that NNs process is hierarchically structured and NNs are only operating in a rather small subset of the space, has been more or less a folklore by the researchers in the neural network community. However, the wide recognition of hierarchy has come late, mostly because the seminal work by Krizhevsky et al. (2012) that proves the significance of hierarchy in NNs experimentally. The hierarchy is mostly motivated by the imitation of biological neural networks (Fukushima (1980) Riesenhuber & Poggio (1999) Riesenhuber & Poggio (2000)), where neuroscience shows that it has a hierarchical organization (Kruger et al. (2013)), and does not make the connection to the hierarchy in nature, which is reasonable since at the time NNs/Perceptron (Rosenblatt (1958)) was invented, the Complex System (Simon (1962) Amderson (1972)) that studies the hierarchy in nature did not exist yet. The connection between hierarchy in nature and NNs has been discussed qualitatively by physicists (Lin & Tegmark (2017) Mehta & Schwab (2014)), though to the best of our knowledge, a fully measure-theoretical characterization of the hierarchy in the data space, described in section 3.1 does not exist before. It gives a theoretical motivation of a hierarchically built hypothesis space, i.e., S-System, contrary to the motivation of artificial NNs, which is an imitation.

### H.2  HIERARCHICAL HYPOTHESIS SPACE OF NNS

Many works have been studying the hierarchical structure of the hypothesis space of NNs. Though perhaps surprisingly, an informal idea similar with S-System has been underlying the design of CNN (Lecun et al. (1998)) at the beginning, where in the unpublished report Bottou et al. (1996), they describe that it is better to defer normalization as much as possible since it "delimiting a priori the set of outcomes", and pass scores as unnormalized log-probabilities. However, perhaps due to a lack of rigor, they removed the discussion in the formal publication. The passing of scores corresponds to the deterministic coupling that transports true measure in the PPMS, while normalization corresponds to assuming a probability kernel to approximate the true measure transported.

Further analysis on the hierarchical behavior of NNs waited for two decades. Early pioneers analyzes from the perspective of kernel space and harmonics. At the end of the dominant era of support vector machine (SVM), Smale et al. (2009) seeks to give NNs a theoretical foundation in Reproducible Kernel Hilbert Space (RKHS) (Vapnik (1999) Scholkopf & Smola (2001)), which is an analogy but may only give limited insights. We will discuss how RKHS relates to S-System later when we discuss the difference between S-System and RKHS based nonlinear algorithms. Many works in this direction have been done, either taking NNs as a recursively built RKHS (Daniely et al. (2016)), or applying the recursion idea to existing kernel methods (Mairal et al. (2014)). We do not aim to cover all kernel works. We envision it as a tool to aid analysis, and design probability kernels in S-System, yet not as the fundamental underpinning. A work (Anselmi et al. (2016)) in the line of RKHS has also sought foundation in probability measure theory, though its focus is the invariance and selectivity of the one layer representation built by NNs. It studies the measure transport due to compact group transformations, and points out that the output of the activation function of NNs could be the probability distribution of low dimensional projection of the measure of data and its transformations, which is similar to the case where S-System only couples group indicator r.e. — they both analyze the grouping of measure transported by transport maps — though when taking on the hierarchical behavior, it falls back to RKHS, and think recursion as "distributions on distributions" instead of coarse grained probability coupling. We believe the work could be inspirational to further refined analysis on r.e.s created by S-System. Under the umbrella of computational harmonics, Mallat (2012) Mallat (2016) understand NNs as a technique that learns a low dimensional function $\Phi(x), x \in \mathcal{X}$ that linearizes the function $f(x)$ to approximate on complex hierarchical symmetry groups from a high dimensional domain $\mathcal{X}$. It achieves this by progressively contracting space volume and linearizing transformation that consists of groups of local symmetries layer by layer. However, the group formalism used is an analogy that only rigorously characterizes Scattering Network (Mallat (2012)), a hierarchical hypothesis space simplified from NNs, and does not characterizes NNs. The group formalism is referred as the "mathematical ghost" in Mallat (2016). We believe these works are important to further incorporate symmetry structure in nature in S-System in future works.

More recently, Ankit B. Patel et al. (2016) interprets NNs in Probabilistic Graphical Model (PGM). It takes activation as log-probabilities that propagate in the net. As the description suggests, it confuses the transported measure to be approximated, and the approximated probability obtained by a probability kernel. Thus, it has to rely on the Gaussian assumption to justify the interpretation, of which the mean serves as templates, and the noise free assumption to justify ReLU activation function. Also, the assumption makes it a generative model that has to make assumptions on the data distribution, while an S-system is able to only make assumptions on how measure is supposed to group. From the spline theory perspective, Balestriero & Baraniuk (2018) understands NNs as a composition of max-affine spline operators, which implies NNs construct a set of signal-dependent, class-specific templates against which the signal is compared via an inner product. From S-System point of view, it is an analysis on the functional form of coupled r.e.s of an S-system that assumes compositional exponential probability kernels and does maximal estimation on group indicator r.e.s. It connects more with the function approximation results, that takes "signal-dependent" as a fact to see what that implies, than the goal of S-System, i.e., giving a theoretical formal definition and interpretation to NNs. We think it may contribute to the refined analysis of decision boundaries in S-System in the future. Analogizing with statistical mechanics, Trevisanutto (2018) takes the group indicator r.e.s. with binary values as gates, of which the expectation will multiply with the coupled r.e.s. to decide how much the "computation" done should be passed on to next layers. However, what is being computed is left unspecified. As in the definition of S-System, the computation is to extend the probability measure space of the measurement collection r.e. that aims to approximate probability measure of events in the event space of PPMS. The group indicator r.e.s. is not a gate, but serves to group measure. It behaves like a gate when its value is binary, yet underlying it serves to create further coupling of grouped measure. Thus, the analog does not unveil the deeper principles underlying, e.g., probability measure space extension and the probability estimation/learning happening in S-System (refer to section 5 appendix G).

### H.3 Machine Learning Algorithm Paradigm

We envision S-System as an attempt that tries to investigate a measure-theoretical foundation of algorithmic modeling methods (Breiman (2001)) for designing machine learning algorithms. Now we

can see NNs as an implementation of S-System, which is a way to transport, group and approximate probability measure. From S-System, we can see that we do not need to make assumptions on the distribution of data to justify that our model is probabilistic — the randomness comes from the data source itself, and it is the probability measure space that a model is manipulating, not the probability values. Thus, we can break from statistics methods developed ever since Ronald Fisher that has to make assumptions on data, and proceed from there. This measure manipulation paradigm may be a promising candidate to the theoretical issue facing high dimensional data analysis (Donoho (2000)). Thus, we discuss current major algorithm paradigms in machine learning/high dimensional data analysis, i.e., Support Vector Machine with Kernels (SVMK) and Probabilistic Graphical Model (PGM).

It is well known that SVMK can be analogized to a NN with one hidden layer. The hypothesis $f$ of SVM can be expressed as a linear combination of inner product between test samples $x_i$ and support vectors $f(\cdot) = \sum_i \alpha_i k(\cdot, x_i)$, where $k$ is the kernel function, and $\alpha_i$ scalars. Writing $f$ in the form of $f(X) = \sum_i \alpha_i k(X, x_i)$, it can be seen that the hypothesis is actually a deterministic coupling, where $f$ is the transport map. As happening in section 5, the training of SVM is also minimizing a surrogate risk between the true data probability measure and the transported measure, though no probability distributions are ever introduced. The probability kernels in S-System is replaced by a positive semi-definite (PSD) kernel, whose output value is a real number indicating something similar with the coupled probability measure of S-System. This observation may seem surprising, however, it makes much sense when we notice the fact that probability is just a function. SVM is a function approximation techniques designed specifically for the case where the data are of high dimensional, yet the number of samples available is small. To combat the curse of dimensionality, it uses a PSD integral operator (Aronszajn (1950)) that maps the sample to a high dimensional space, which can be taken as templates, and only approximates measure that is in the vicinity of those templates and ignores the rest of the space. The kernel can also be built hierarchically, which is discussed in appendix H.2. For the time being, S-System does not contain SVMK as a special case, while we envision by properly generalizing the probability kernels in CGE, a large class of algorithms may include SVM.

As for PGM, it is a special case of S-System. As mentioned repeatedly throughout the paper, S-System merely makes assumptions on how measure is supposed to group, without making assumptions on the actual distribution of the data. The learning framework of S-System described in appendix G is actually the same as PGM when only considering the unsupervised case, where assumptions on data distribution have to be made. Thus, S-System is a superset of algorithms including PGM. The graph in PGM is actually a poset. However, the insight comes from where they differ. Relying heavily on the assumptions on the distribution of data, which is in reality unknown, it introduces large model biases, which perhaps is the reason why it alone cannot compete with NNs on complex high dimensional data. Furthermore, S-System is naturally compatible with supervised labels, since hidden variables/group indicator r.e.s map one-to-one to labels, which dictates how measure should be grouped. This point is discussed more thoroughly in appendix G, where supervised and unsupervised learning are taken as dual perspectives on the same object.

## H.4 Geometry

In the related works on hierarchical hypothesis space discussed earlier, all of them have their own geometry, we only discuss related works in this subsection that are related to the geometry defined in section 4.2.

Most of the works we are aware of that try to endow a geometry on NNs through information geometry (IG) were done before the deep learning era, not surprisingly, by Amari, who developed IG. All the works study NNs with a single hidden layer. Amari & Nagaoka (2007) formulates the manifold parameterized by all parameters of a NN as neuromanifold, while in section 4.2, the manifold we defined focuses on the submanifolds indexed by a scale poset, which will be discussed more in the next paragraph. Actually, the neuromanifold is the stochastic manifold consisting of possible probability measure on the random element set of the event representation built by S-System. Two directions of analysis have been made. The first is to study the behavior of the curved exponential families obtained by conditioning, which is done in Amari (1995), and falls in the category of generative training. The other is to study supervised trained NNs, and study the neuromanifold, with a focus on the impact of singularities on training dynamics (Amari et al. (2006)). The later proposed the

Natural Gradient Descent methods, and many works have been working on it thereafter, which we will not discuss. The study on the hierarchy has been limited on decomposition of high order interactions in a single hidden layer NN (Amari (2001)) without attacking the recursion in NNs, though we tend to think NNs with more layers unroll higher order interactions, but we do not find that they pursue this path. As mentioned, the study of hierarchical behaviors of NNs has been absent, which is the focus of this paper, and is emphasized in the paragraph below.

The geometry defined in section 4.2 is to investigate the hierarchical geometry of NNs. The compositional exponential family gives the definition of Neural Network Manifolds that properly identifies the curved exponential families, or in other words, submanifolds, in a probability family built by the overall parameter space of NNs, which is complicated, e.g., containing singularities (Amari et al. (2006)). Note that we do not differentiate submanifolds with manifolds in the main content to avoid clutters. As discussed, the submanifolds are well represented by their expectation statistics, and the definition identifies how coarse graining in divergence happens in theorem 4.1. Thus, definitions given are distinctive in characterizing the hierarchical geometry of NNs, which is absent in previous works, which either stay in the realm of single hidden layer (Amari (1995) Amari (2001)), or take the whole parameter space as the parameterization of a manifold that contains singularities (which rigorously is not a manifold) (Amari et al. (2006)), though we are well aware that the works present in this paper are merely scratching the surface. Our focus for now is merely to show the coarse graining contraction effect of CGE quantitatively, and much more works are to be done, e.g., the hierarchical and within-layer interactions between these submanifolds. As a concrete example, it is known that the EM algorithm has an IG interpretation (Amari (1995)). The expectation, KL divergence minimization interpretation of the back propagation algorithm in appendix G can be interpreted similarly from the IG perspective. Thus, Amari (1995) can be generalized to NNs with arbitrary number of layers, and in generative or supervised training settings. It implies the two directions mentioned in the previous paragraph can be unified, though further analysis on its impact, e.g., the analysis of singularities, needs more works.

Very recently, at the time of writing this paper, a few reports have been submitted on the archive that try to attack the supervised deep NNs (Amari et al. (2018a) Amari et al. (2018b)). But again, they follow their old idea that analyzes the whole neuromanifold. It assumes weights and biases of NNs to be Gaussian, and study how properties related to the distribution of activations of each layer change, e.g., fisher information matrix, without trying to formally define the geometry, or the submanifold structure in the intermediate layers of NNs.

Lastly, we note that Lin & Tegmark (2017) also tries to discuss the coarse graining effect in term of the information monotony phenomenon as "information distillation", but it does that rather generally and qualitatively, does not put the phenomenon in an exact NN context, and not make the connection between it and the geometry in information.

### H.5 LEARNING AND OPTIMIZATION

We cover related works on learning and optimization of NNs and S-System in this subsection.

The learning framework is an application of a general probability estimation framework on the particular case of S-System, thus, the reader may find the learning framework similar with variational inference widely used in existing probabilistic graphical models. However, the similarity lies in the fact that both S-System and PGM approximate probability, of which the decomposition of complete data likelihood is about probability to be estimated, not about specific hypotheses in use. The difference between S-System and PGM has been detailed in appendix H.3. Previously, the BP algorithms have mostly been viewed as a heuristic tool, instead of having a theoretically rigorous derivation. The learning framework of S-System shows that the FP and BP are actually maximizing the complete data likelihood, and are not merely minimizing the discrepancy between the estimated conditional probability of labels given data with the true conditional probability through empirical risk minimization, but also maximizing the expected data likelihood through activation function.

Similar to the study on the hierarchical hypothesis of NNs, the study on optimization gains its moment rather recently. We focus on the works that attack the full complexity of optimization problem of deep NNs, while for more related works, we refer the readers to related works discussed in Dauphin et al. (2014) Nguyen & Hein (2017) Liang et al. (2018) for works before the deep learning era, on shallow networks and NP-hardness of NN optimization.

Roughly, two approaches have been taken in analyzing the optimization of NNs, one from the linear algebra perspective, the other from mean field theory using random matrix theory. Our work falls in the latter approach. The linear algebra approach, as the name suggests, shies away from the nonlinear nature of the problem. Kawaguchi (2016) proves all local minima of a deep linear NN are global minima when some rank conditions of the weight matrices are held. Nguyen & Hein (2017) Nguyen & Hein (2018) prove that if in a certain layer of a NN, it has more neurons than training samples, which makes it possible that the feature maps of all samples are linearly independent, then the network can reach zero training errors. A few works following in the linear-algebraic direction (Laurent & von Brecht (2018) Liang et al. (2018) Yun et al. (2018)) improve upon the two previous results, but using essentially the same approach. As the conditions in Kawaguchi (2016) indicate, the rank related linear algebraic condition does not transport to nonlinear NNs. While for Nguyen & Hein (2017), it characterizes a phenomenon that if in a layer of a NN, it can allocate a neuron to memorize each training sample, then based on the memorization, it can reach zero errors. In a certain way, we believe NNs are doing certain memorization, for the fact that the output elements in the intermediate event representations are learning template/mean of events, as discussed in section 4.2. However, it does it in a smart way, where the templates are decomposed hierarchically. Thus, it is likely we do not need so many linearly independent intermediate features, which would lead to poor generalization. Thus, to truly understand the optimization behavior of NNs, we need to step out of the comfort zone of linearity.

The mean field theory approach using the tools of random matrix theory can attack the optimization of NNs in its full complexity, though existing works tend to be confused on the source of randomness. Due to an inadequate understanding of the randomness induced by activation function, Choromanska et al. (2015a) tries to get rid of the group indicator r.e.s. by assuming that its value is independent of the input r.e.s. of CGE, which is unrealistic (Choromanska et al. (2015b)), nevertheless it is a brave attempt, and the first paper to attack a deep NN in its full complexity. After Choromanska et al. (2015a) which approaches by analogizing with spin glass systems — it is a complex system, as NNs are — some researchers start to study NNs from mean field theory from the first principle instead of by analog. Again, confused with the source of randomness in activation in the intermediate layers of NNs, Jeffrey Pennington (2017) just assumes data, weights and errors are of i.i.d. Gaussian distribution, which are mean field approach assumptions and unrealistic, and proceeds to analyze the Hessian of the loss function of NNs, though due to limitations of their assumptions, they can only analyze a NN with one hidden layer. By laying a theoretical foundation of NNs, S-System accurately points out where randomness arises in NNs, and what reminds to prove theorem 5.1 is to find the right random matrix tools.

