# OpenReview forum: "S-System, Geometry, Learning, and Optimization: A Theory of Neural Networks"
_ICLR.cc/2019/Conference_

### Official Review · AnonReviewer2 · 2018-11-07
**has grand ideas but poorly written, cannot check for correctness**

**Rating:** 4
**Confidence:** 1

**Review:**

The paper is extremely difficult to read. There are too many concepts introduced at once, casual comments mixed with semi-formal statements. The theorems sound interesting, the implications are grand and of interest to ICLR, but the proofs are impossible to follow. As such, I am not in a position to make a recommendation.

I strongly recommend the authors to split the paper into multiple parts with clear-cut statements in each, with clear and detailed proofs, and submit to appropriate journals / conferences.

---

### Official Review · AnonReviewer4 · 2018-11-11
**Interesting take on neural networks from a measure-theoretic viewpoint, however not easy to follow for a non-expert**

**Rating:** 4
**Confidence:** 2

**Review:**

The paper provides a new framework "S-System" as a generalization of hierarchal models including neural networks. The paper shows an alternative way to derive the activation functions commonly used in practice in a principled way. It further shows that the landscape of the optimization problem of neural networks has nice properties in the setting where the number of input/hidden units tending to infinity and the neurons satisfy certain diversity conditions.

Overall, the paper presents super interesting ideas that can potentially lead to a deeper understanding of the fundamentals of deep learning. However, for a general reader it is a hard-to-follow paper. Without a full understanding of the various domains this paper presents ideas from, it is hard to verify and fully understand the claims. I believe the paper would be better appreciated by an audience of a mathematical journal. As an alternative, I would encourage the readers to split the paper and possibly simplify the content by using a running example (more concrete than the one of MLP used) to explain the implications as well as assumptions.

A clearer, more accessible presentation is necessary so that a non-expert can understand the paper's results. Thus, I vote to reject.

---

### Meta-Review · Area_Chair1 · 2018-12-14
**Paper unreadable**

**Confidence:** 5
**Recommendation:** Reject

**Metareview:**

The paper is extremely difficult to read, even given that both reviewers have very strong math / theoretical background. Although it may potentially include interesting ideas, nothing in the work could not be understood by the ICLR audience.